# Distributed Online Convex Optimization with Compressed Communication

**Zhipeng Tu**[1,2], **Xi Wang**[1,2], **Yiguang Hong**[*3], **Lei Wang**[4], **Deming Yuan**[5], **Guodong Shi**[2]

[1]Academy of Mathematics and Systems Science, Chinese Academy of Sciences, China
[2]Australian Center for Field Robotics, School of AMME, The University of Sydney, Australia
[3]Department of Control Science and Engineering, Tongji University, China
[4]College of Control Science and Engineering, Zhejiang University, China
[5]School of Automation, Nanjing University of Science and Technology, China
`tuzhipeng@amss.ac.cn, wangxi14@mails.ucas.ac.cn, yghong@iss.ac.cn`
`lei.wangzju@zju.edu.cn, dmyuan1012@gmail.com, guodong.shi@sydney.edu.au`

## Abstract

We consider a distributed online convex optimization problem when streaming data are distributed among computing agents over a connected communication network. Since the data are high-dimensional or the network is large-scale, communication load can be a bottleneck for the efficiency of distributed algorithms. To tackle this bottleneck, we apply the state-of-art data compression scheme to the fundamental GD-based distributed online algorithms. Three algorithms with difference-compressed communication are proposed for full information feedback (DC-DOGD), one-point bandit feedback (DC-DOBD), and two-point bandit feedback (DC-DO2BD), respectively. We obtain regret bounds explicitly in terms of time horizon, compression ratio, decision dimension, agent number, and network parameters. Our algorithms are proved to be no-regret and match the same regret bounds, w.r.t. time horizon, with their uncompressed versions for both convex and strongly convex losses. Numerical experiments are given to validate the theoretical findings and illustrate that the proposed algorithms can effectively reduce the total transmitted bits for distributed online training compared with the uncompressed baseline.

## 1 Introduction

Online optimization has attracted considerable attention in recent decades, for its remarkable applications in machine learning tasks such as spam filtering, dictionary learning, advertising selection, and so on [1, 2, 3]. In such online tasks, data are revealed incrementally, and decisions must be made before all data are available. When the streaming data are collected at multiple agents, the distributed online optimization over a multi-agent network is considered, where data storage and processing are performed in the agents [4, 5]. It is often impractical to communicate data among different agents from multiple concerns such as privacy and bandwidth utilization. Also, there is no central agent for global coordination. In such settings, each agent relies on its own data to run an algorithm while communicating decisions with its immediate neighbors.

To be specific, this paper considers the distributed online convex optimization (DOCO) problem over an $N$-agent network. The objective is to minimize the accumulated system-wide loss $\min_{x \in \mathcal{K}} \sum_{t=1}^{T} \sum_{i=1}^{N} f_i^t(x)$, where the local convex loss function $f_i^t$ is formed by the data arriving

---

[*]Corresponding author

36th Conference on Neural Information Processing Systems (NeurIPS 2022).

at time $t$ in agent $i$, and $\mathcal{K} \subset \mathbb{R}^d$ is a convex feasible set. Note that the loss information is revealed to agent $i$ after its decision $x_i^t$ is made. Generally, there are two basic types of information feedback that agents can possess. One is the full information feedback, where agents have access to the loss functions. The other is the bandit feedback, where agents can only possess the values of the loss function at points around the decision. At each time step, agents choose the decisions based on their local feedback and neighbors' information. To measure the performance of an algorithm, the (static) regret is frequently used, which compares the cumulative loss of online decisions and the loss of the best decision chosen in hindsight through all the time horizons. The regret of node $j \in \mathcal{V}$ is defined as

$$\mathrm{R}(j, T) = \sum_{t=1}^{T} \sum_{i=1}^{N} f_i^t(x_j^t) - \min_{x \in \mathcal{K}} \sum_{t=1}^{T} \sum_{i=1}^{N} f_i^t(x). \tag{1}$$

An algorithm is called *no-regret* [6] if the average regret over $T$ goes to zero, which means that the online decision updated by the streaming data is not far from the best decision chosen in hindsight. Distributed no-regret online algorithms have been widely studied in recent years [4, 7, 8, 9, 10, 11].

Although distributed algorithms are theoretically feasible, most of them are not practical as the model size gets large, since communication cost can be a bottleneck for efficiency. In distributed training tasks, agents can be powful microcomputers, while their communication network may be with low bandwidth. The information exchange over the network is pretty slow compared with the computation taking place in agents [12]. Thus, communication compression techniques are of significance for practical implementations.

There have been many attempts to combine distributed optimization algorithms with compressors. A straightforward idea is the direct compression scheme, while algorithms with this simple scheme fail to converge even for the distributed average consensus problem [13, 14]. As an improvement, extrapolation compression scheme and difference compression scheme are proposed [15]. Along this line, quite a number of studies successfully extend the distributed optimization algorithms with compressors and meanwhile maintain the convergence rate [16, 17, 18, 19]. However, distributed online optimization with compression is still an area that has not been fully exploited. [20] proposed ECD-AMSGrad algorithm that extended the AMSGrad to the distributed online setting with extrapolation compression, while only empirical results were given without theoretical analysis. The key open problem in this area is

> *whether it is possible to design provably no-regret distributed online algorithms that work with compressors.*

**Contributions** In this work, we answer the above question in the affirmative. We apply the difference compression scheme to the fundamental GD-based distributed online algorithms. Although the idea of such combination is simple, the underlying algorithm design and theoretical principle are challenging since the compression error, projection error, and consensus error will be coupled. Our contributions are summarized as follows:

- We propose communication-efficient distributed online algorithms, which consist of difference compression, $\gamma$-gossip consensus, gradient descent, and projection, for the cases of full information feedback (DC-DOGD), one-point bandit feedback (DC-DOBD), and two-point bandit feedback (DC-DO2BD), respectively. We make the technical advance to combine the difference compression scheme with the projection scheme. Through proper design, the errors can be estimated and controlled with the consensus stepsize $\gamma$ and the gradient descent stepsizes.

- We analyze the regret bounds of the proposed algorithms for convex and strongly convex losses, respectively, which are established explicitly in terms of time horizon $T$, compression ratio $\omega$, decision dimension $d$, agent number $N$, and parameters of the communication graph $\mathcal{G}$, as simplified and summarized in Table 1. The obtained regret bounds are in accordance with those of [11] w.r.t $T, N, d$. Our algorithms are no-regret with theoretical guarantees.

- We give exhaustive experiments to illustrate the performance of the proposed algorithms. Compared with the uncompressed algorithm DAOL [8], the proposed algorithms can reduce the total transmitted bits for distributed online training. Moreover, DC-DOGD and DC-DO2BD significantly outperform the algorithm ECD-AMSGrad [20].

Table 1: Regret bounds in different settings

| Settings | convex losses | strongly convex losses |
|---|---|---|
| Full information | $\mathcal{O}\left(\left(\omega^{-2}N^{1/2}+\omega^{-4}\right)N\sqrt{T}\right)$ | $\mathcal{O}\left(\left(\omega^{-2}N^{1/2}+\omega^{-4}\right)N\ln(T)\right)$ |
| One-point bandit | $\mathcal{O}\left(\left(\omega^{-2}N^{1/2}+\omega^{-4}\right)^{1/2}Nd^{1/2}T^{3/4}\right)$ | $\mathcal{O}\left(\left(\omega^{-2}N^{1/2}+\omega^{-4}\right)^{1/3}Nd^{2/3}T^{2/3}\ln^{1/3}(T)\right)$ |
| Two-point bandit | $\mathcal{O}\left(\left(\omega^{-2}N^{1/2}+\omega^{-4}\right)Nd\sqrt{T}\right)$ | $\mathcal{O}\left(\left(\omega^{-2}N^{1/2}+\omega^{-4}\right)Nd^2\ln(T)\right)$ |

**Related Work**  Distributed online convex optimization has received numerous attention in recent years. Many basic algorithms have been extended to distributed settings, referring to [7] and references therein. For instance, [8] proposed a distributed online subgradient algorithm over a static directed network and achieved the regrets $\mathcal{O}(\sqrt{T})$ and $\mathcal{O}(\ln(T))$ for convex and strongly convex losses, which are in line with those of centralized online algorithms [21, 22]. In the bandit feedback setting, by modifying the gradient to a randomized estimator, [9] proposed a distributed online bandit algorithm with one-point sampling over an undirected network and achieved $\mathcal{O}(T^{2/3}\ln^{1/3}(T))$ regret for strongly convex losses. Better bound could be achieved by two-point sampling, as [4] obtained $\mathcal{O}(\sqrt{T})$ regret for convex quadratic losses in case of bounded decision set. Also, [10] investigated the distributed online two-point bandit algorithm in dynamic environments and achieved $\mathcal{O}(\sqrt{T})$ and $\mathcal{O}(\ln(T))$ regrets for convex and strongly convex losses, which match those of centralized two-bandit algorithms [23]. [11] comprehensively studied DOCO over Erdős-Rényi random networks in full gradient feedback, one-point bandit feedback, and two-point bandit feedback, and gave regret bounds. Along the line of [11], this paper aims to further introduce compressed communication strategies, while preserving the regret bounds order-wisely.

Recently, combining distributed optimization algorithms with compressors has seen a dramatic rise in interest. Traditional compressors include the quantization [24, 25, 26], sparsification [27, 28, 29], and hybrid combining of them [30, 31, 32]. The way to apply compressors is called a compression scheme. The most widely used compression schemes in distributed optimization are extrapolation compression and difference compression [15]. Extrapolation compression allows agents to compress the extrapolation between the last two local states. Decentralized PSGD with extrapolation compression (ECD-PSGD) [15] was proved to converge sublinearly and match the rate of its uncompressed case (D-PSGD). Difference compression (DC), which is also called CHOCO [16] or innovation compression [19], allows agents to add replicas of neighboring states and compress the state-difference. There have been extensive successful designs combining distribute optimization algorithms with DC, to name a few, DCD-PSGD [15] (based on PSGD), CHOCO-SGD [16] (based on gossip SGD), SPARQ-SGD [17] (based on event-trigger), C-GT [18] (based on gradient tracking), and COLD[19] (based on NIDS), etc.

The difference compression scheme is different from the error-feedback scheme (EF) [24, 33] in what to be compressed, which results in different application fields. DC compresses the difference between the current variable and the replica variable, which is widely used in distributed optimization where nodes tend to exchange state information whose limit is nonzero in general. EF compresses the sum of the gradient and the residual error, which is widely used in federated learning where nodes tend to exchange gradient information whose limit is expected to be zero. The insight is that successful designs have to compress something that goes to zero, otherwise the noise introduced by the compression will not vanish, which leads to the algorithm oscillation or even divergence. It is worth noting that the idea of DC can also be applied to federated learning, such as EF21 [34] and FedPAQ [35], which is another line of research. In this paper, we focus on applying DC to distributed online optimization.

The results about the distributed online optimization with compression are quite limited. [20] poposed the ECD-AMSGrad algorithm that combined AMSGrad with extrapolation compression. Actually, the AMSGrad algorithm may not be a good choice for DOCO problem since although AMSGrad itself is proved no-regret [36], a considerable performance gap still exists between AMSGrad and SGD [37]. Besides, the introduction of compression errors will further worsen the algorithm such that ECD-AMSGrad will lose the no-regret performance (see Section 5). This paper focuses on the fundamental GD-based algorithms and will give no-regret guarantees.

## 2 Full Information Feedback

In this section, we first introduce the multi-agent network and the compressor we use, and then propose a communication-efficient distributed online algorithm for DOCO with full information feedback. Expected regret bounds will be given for both convex and strongly convex losses.

**Graph** The multi-agent network is described by an undirected graph $\mathcal{G}(\mathcal{V}, \mathcal{E})$, where $\mathcal{V} = \{1, \ldots, N\}$ is the set of nodes, representing the set of agents, and $\mathcal{E} \subset \mathcal{V} \times \mathcal{V}$ is the set of edges. Let $A = [a_{ij}] \in \mathbb{R}^{N \times N}$ be the connectivity matrix of $\mathcal{G}$ such that $a_{ij} = a_{ji}$. If $(v_i, v_j) \in \mathcal{E}$, then $v_i$ and $v_j$ can exchange information, and $a_{ij} = 0$ otherwise. It is worth noting that the communication is node-to-node in our distributed setting, and there is no central node. The graph in this paper satisfies the following assumption.

**Assumption 1.** *The communication graph $\mathcal{G}$ is undirected and connected. Its connectivity matrix $A \in [0, 1]^{N \times N}$ is a symmetric doubly stochastic matrix.*

**Compressor** A compressor $Q(\cdot) : \mathbb{R}^d \to \mathbb{R}^d$ is a mapping whose output can be usually encoded with fewer bits than its input. In this paper, we consider a broad class of compressors with the following general property, which has been widely considered in distributed optimization with compression [16, 17, 19].

**Assumption 2.** *For some $\omega \in (0, 1]$, $Q$ satisfies*

$$\mathbb{E}_Q \|Q(x) - x\|^2 \leq (1 - \omega)\|x\|^2, \quad \forall x \in \mathbb{R}^d, \tag{2}$$

*where $\mathbb{E}_Q$ denotes the expectation over the internal randomness of $Q$.*

Compressors satisfy the above assumption are called $\omega$-contracted, which include many important biased or unbiased compressors, such as sparsification ($\mathrm{Rand}_k$ and $\mathrm{Top}_k$) [28], random quantization ($\mathrm{QSGD}_s$) [26], random gossip ($\mathrm{RGossip}_p$) [16], etc.

### 2.1 Algorithm design

In the full information feedback, the loss function $f_i^t$ is revealed to node $i$ at time $t$ after the decision $x_i^t$ is made. Then node $i$ has access to the gradient value $\nabla f_i^t(x_i^t)$ and can use $g_i^t = \nabla f_i^t(x_i^t)$ to make the next decision $x_i^{t+1}$. We propose the DC-DOGD algorithm as shown in Algorithm 1, which is based on DAOL [8] and memory-efficient CHOCO-SGD [16]. The DC-DOGD algorithm consists of two main parts: difference compressed communication (steps 2 and 3) and local decision update (steps 4 and 5).

---

**Algorithm 1** Distributed Online Gradient Descent with Difference Compression (DC-DOGD)

---

**Input:** consensus stepsize $\gamma$, GD stepsizes $\{\eta_t\}_{t=1}^T$, time $T$
**Initialize:** set $x_i^1 = \mathbf{0}, \hat{x}_i^1 = \mathbf{0}, s_i^1 = \mathbf{0}$, for each node $i \in \mathcal{V}$.
1: **for** $t = 1$ to $T - 1$ **do** in parallel for each node $i \in \mathcal{V}$
2:     Compress the difference vector $q_i^t = Q(x_i^t - \hat{x}_i^t)$ and update the replica $\hat{x}_i^{t+1} = \hat{x}_i^t + q_i^t$.
3:     Send $q_i^t$ to its neighbors and receive $q_j^t$ from all its neighbors $j \in \mathcal{N}_i$. Update the estimate of the consensus decision by $s_i^{t+1} = s_i^t + \sum_{j \in \mathcal{N}_i} a_{ij} q_j^t$.
4:     Receive the full information feedback and calculate $g_i^t = \nabla f_i^t(x_i^t)$.
5:     Update its decision variable as follows

$$x_i^{t+1} = P_{\mathcal{K}}\left(x_i^t + \gamma(s_i^{t+1} - \hat{x}_i^{t+1}) - \eta_t g_i^t\right), \tag{3}$$

    where $P_{\mathcal{K}}$ denotes the Euclidean projection, i.e., $P_{\mathcal{K}}(x) = \mathrm{argmin}_{y \in \mathcal{K}} \|x - y\|$.
**Output:** $\{x_i^t\}_{t=1}^T$

---

The insight of introducing the variable $\hat{x}_i^t$ and the difference compression are as below. Assume that $x^* \neq 0$ without loss of generality. Then, if node $i$ transmits the directly compressed information $Q(x_i^t)$ to its neighbors, the compression error $Q(x_i^t) - x_i^t$ will not vanish for $t \to \infty$. The accumulation of compression errors makes the algorithm fail to converge. Instead, we compress something that

goes to zero. Let node $i$ and all its neighbors keep an auxiliary variable $\hat{x}_i^t$ locally, which acts as a replica of $x_i^t$. Whenever node $i$ updates its decision variable $x_i^t$, node $i$ calculates the difference $x_i^t - \hat{x}_i^t$, compresses the difference $q_i^t = Q(x_i^t - \hat{x}_i^t)$, and sends the compressed information $q_i^t$ to its neighbors. After that, node $i$ and all its neighbors update the local replica $\hat{x}_i^{t+1} = \hat{x}_i^t + q_i^t$. When all nodes are reaching a consensus optimal decision, the updates of local decisions are small, and the differences between the replica variables and the true decision variables are also small. Then the compression errors are expected to vanish.

Local decision variables update through gradient descent, $\gamma$-gossip, and projection, in order to minimize the local loss function, keep consensus with neighbors, and remain in the feasible set, respectively. For each node $i$, $s_i^1$ is initialized to $\mathbf{0}$, and thus, $s_i^t = \sum_{j=1}^{N} a_{ij}\hat{x}_j^t$. Recall that $\hat{x}_i^t$ tracks $x_i^t$, then $s_i^t$ acts as node $i$'s estimate of the consensus decision at time $t$. The $\gamma$-gossip protocol is adopted to renovate the decision variable towards the consensus decision. The consensus stepsize $\gamma \in (0, 1]$ is tunable to control the consensus speed, and will also play a crucial role in controlling the compression error.

If there is no compression, i.e., using exact communication, then $\hat{x}_i^{t+1}$ turns out to be $x_i^t$, and $s_i^{t+1}$ becomes $\sum_{j=1}^{N} a_{ij}x_j^t$. Besides, take $\gamma = 1$, and then (3) reduces to

$$x_i^{t+1} = P_{\mathcal{K}}\left(x_i^t + \gamma \sum_{j \in \mathcal{N}_i} a_{ij}(x_j^t - x_i^t) - \eta_t \nabla f_i^t(x_i^t)\right) = P_{\mathcal{K}}\left(\sum_{j \in \mathcal{N}_i} a_{ij}x_j^t - \eta_t \nabla f_i^t(x_i^t)\right),$$

which is the DAOL algorithm in [8].

**Remark 1.** $s_i^t$ *is introduced for the memory-efficiency. Eq. (3) is equivalent to the update rule*

$$x_i^{t+1} = P_{\mathcal{K}}\left(x_i^t + \gamma \sum_{j \in \mathcal{N}_i} a_{ij}(\hat{x}_j^{t+1} - \hat{x}_i^{t+1}) - \eta_t g_i^t\right). \tag{4}$$

*If we adopt the update rule (4) together with $\hat{x}_j^{t+1} = \hat{x}_j^t + q_j^t$ for $j \in \mathcal{N}_i \cup \{i\}$ instead of steps 3 and 5, then each node have to store $deg(i) + 2$ vectors, namely, $x_i, \hat{x}_i$ and $\hat{x}_j, j \in \mathcal{N}_i$, which is memory-consuming.*

**Remark 2.** *It is potential to extend our algorithm from a fixed graph to a time-varying graph $\mathcal{G}_t = (\mathcal{V}, \mathcal{E}_t)$. We give the following two settings.*

- *Random switching undirected networks. Our algorithm can be directly applied to the classic time-varying Erdős-Rényi random networks [38], where $\mathcal{G}_t$ is generated over the prescribed graph $\mathcal{G}$ and $\{i, j\} \in \mathcal{E}_t$ with a probability $0 < p < 1$ for all $\{i, j\} \in \mathcal{E}$. This setting is equivalent to performing random gossip compressor over the fixed graph $\mathcal{G}$, that is, transmitting information with the probability $p$, which satisfies Assumption 2 with $\omega = p$.*

- *Deterministic switching undirected networks. Our algorithm can be applied to this setting by modifying the connectivity matrix from $A$ to $A(t)$. With further assumptions on the switching networks such as Bounded Intercommunication Interval assumption [39], one can analyze the regret interval-wisely. This is interesting but out of this paper's focus, and we leave it for future study.*

## 2.2 Regret bounds

We consider following assumptions, which are widely used in the studies of distributed online optimization [1, 11, 40].

**Assumption 3.** *The convex set $\mathcal{K}$ is bounded with diameter $D$, i.e., $\|x - y\| \leq D, \forall x, y \in \mathcal{K}$.*

**Assumption 4.** *For each $i \in \mathcal{V}$ and $t = 1, 2, ..., T$, the loss function $f_i^t$ is convex with bounded gradient over $\mathcal{K}$, i.e., $\max_{i,t,x} \|\nabla f_i^t(x)\| \leq G$.*

**Assumption 5.** *For each $i \in \mathcal{V}$ and $t = 1, 2, ..., T$, the loss function $f_i^t$ is $\mu$-strongly convex over $\mathcal{K}$ with the parameter $\mu > 0$, i.e., $f_i^t(x) - f_i^t(y) \geq \langle x - y, \nabla f_i^t(y)\rangle + \frac{\mu}{2}\|x - y\|^2, \forall x, y \in \mathcal{K}$.*

Suppose that the eigenvalues of the symmetric doubly stochastic connectivity matrix $A$ are $1 = |\lambda_1(A)| > |\lambda_2(A)| \geq \cdots \geq |\lambda_N(A)|$. Define the spectral gap $\delta := 1 - |\lambda_2(A)| \in (0, 1]$ and the spectral radius of the Laplacian matrix $\beta := \|I_N - A\|_2 \in [0, 2]$. Then we give the expected regret bounds of Algorithm 1 for convex and strongly convex losses, respectively.

**Theorem 1.** *Let Assumptions 1, 2, 3 and 4 hold. Consider Algorithm 1 with the consensus stepsize*

$$\gamma = \frac{3\delta^3\omega^2(\omega+1)}{48(\delta^2+18\delta\beta^2+36\beta^2)\beta^2(\omega+2)(1-\omega)+4\delta^2(\beta^2+\beta)(\omega+2)(1-\omega)\omega+6\delta^3\omega}, \quad (5)$$

*(i) (Convex case) Take the gradient descent stepsize $\eta_t = \frac{D}{G\sqrt{t+c}}$ for a constant $c \geq \frac{8}{3\gamma\delta}$. Then for each $j \in \mathcal{V}$ and $T \geq 1$,*

$$\mathbb{E}_Q\left[\mathrm{R}(j,T)\right] \leq \left(\frac{1}{2} + 8\sqrt{3}\left(\sqrt{N} + 2\sqrt{3}\gamma^{-1}\delta^{-1} + 1\right)\left(1 + \gamma^{-1}\delta^{-1} + \omega^{-1}\right)\right)NGD\sqrt{T+c}. \tag{6}$$

*(ii) (Strongly convex case) With additional Assumption 5, take the gradient descent stepsize $\eta_t = \frac{1}{\mu(t+c)}$ for a constant $c \geq \frac{16}{3\gamma\delta}$. Then for each $j \in \mathcal{V}$ and $T \geq 1$,*

$$\mathbb{E}_Q\left[\mathrm{R}(j,T)\right] \leq \mu c D^2 + 4\sqrt{3}\left(\sqrt{N} + 2\sqrt{3}\gamma^{-1}\delta^{-1} + 1\right)\left(1 + \gamma^{-1}\delta^{-1} + \omega^{-1}\right)NG^2\mu^{-1}\ln(T+c). \tag{7}$$

The proof ideas are as follows. Firstly, we estimate the general regret bounds for each node, which depend on the consensus error, the projection error, the compression error, and the gradient descent stepsize. Then comes the key point that we analyze the coupled relationship between the errors, and bound them with the consensus stepsize $\gamma$ and the GD stepsize $\eta_t$. Finally, we choose proper $\gamma$ and $\eta_t$ to obtain Theorem 1. Complete proofs are attached to Appendix B.

The consensus stepsize $\gamma$ chosen in (5) depends on the compression ratio $\omega$ and the communicaiton graph paremeters $\delta$ and $\beta$. Notice that $\gamma$ is an increasing function with respect to $\omega$, and $\gamma|_{\omega=0} = 0, \gamma|_{\omega=1} = 1$. Thus, $\gamma \in (0,1]$ for $\omega \in (0,1]$. If there is no compression ($\omega = 1$) , then $\gamma = 1$, and Algorithm 1 exactly reduces to DAOL [8], as mentioned in the algorithm design.

Theorem 1 shows that Algorithm 1 achieves the regret bounds $\mathcal{O}((\omega^{-2}N^{1/2} + \omega^{-4})N\sqrt{T})$ and $\mathcal{O}\left(\left(\omega^{-2}N^{1/2} + \omega^{-4}\right)N\ln(T)\right)$ for convex losses and strongly convex losses, respectively. The results suggest that

- Algorithm 1 is no-regret in both convex case and strongly convex case, since the time averaged regret $\mathbb{E}_Q[\mathrm{R}(j,T)]/T \to 0$ for $T \to \infty$. The obtained regret bounds $\mathcal{O}(\sqrt{T})$ and $\mathcal{O}(\ln(T))$ are in accordance with those of the centralized online algorithms in the respective cases [21, 22].

- The node averaged regret $\mathbb{E}_Q[\mathrm{R}(j,T)]/N$ increases with $N$, which in line with the result in [11].

- As the compression ratio $\omega$ decreases, fewer bits are needed for node-to-node communication in each iteration, while more iteration rounds are needed to reach the desired regret. $\omega$ can be used to balance the iteration rounds and the transmitted bits in each iteration from multiple concerns such as the bandwidth and agent computation capability. In practice, we can choose a proper $\omega$ to minimize the total transmitted bits or minimize the overall training time.

## 3 One-point Bandit Feedback

In this section, we apply difference compression to DOCO with one-point bandit feedback. We propose DC-DOBD algorithm, which basically follows DC-DOGD, except for the gradient estimation.

In the one-point bandit feedback, after making the decision $x_i^t$ at time $t$, agent $i$ can query the loss function value at one point around $x_i^t$ and use the feedback to construct the gradient estimator $g_i^t$. Like the procedure in [41], let agent $i$ choose a unit-norm vector $u_i^t \in \mathbb{R}^d$ uniformly at random, query the value of $f_i^t$ at the point $y_i^t = x_i^t + \epsilon u_i^t$, and calculate $g_i^t = \frac{d}{\epsilon}f_i^t(y_i^t)u_i^t$. Since the loss function $f_i^t$ is defined in the set $\mathcal{K}$, we slightly modify the projection in (3) as $P_{(1-\zeta)\mathcal{K}}$ to ensure the query point $y_i^t \in \mathcal{K}$. Algorithm 2 actually performs the gradient descent on the function $\hat{f}_i^t(x) = \mathbb{E}_{u\in\mathcal{B}}\left[f_i^t(x+\epsilon u)\right]$ restricted to the convex set $(1-\zeta)\mathcal{K}$. It has been shown by [41] that $\mathbb{E}\left[g_i^t\right] = \nabla\hat{f}_i^t(x_i^t)$. In the bandit setting, Assumptions 3 and 4 are modified as follows, which are common in online bandit optimization [11, 23, 41].

**Assumption 6.** *The convex set $\mathcal{K}$ contains the ball of radius $r$ centered at the origin, and is contained in the ball of radius $R$, i.e., $r\mathcal{B} \subseteq \mathcal{K} \subseteq R\mathcal{B}$, $\mathcal{B} = \{u \in \mathbb{R}^d : \|u\| \leq 1\}$.*

**Algorithm 2** Distributed Online One-point Bandit Gradient Descent with Difference Compression (DC-DOBD)

---

**Input:** consensus stepsize $\gamma$, GD stepsizes $\{\eta_t\}_{t=1}^T$, time $T$, exploration parameter $\epsilon$, shrinkage parameter $\zeta$

**Initialize:** set $x_i^1 = \mathbf{0}, \hat{x}_i^1 = \mathbf{0}, s_i^1 = \mathbf{0}$, for each node $i \in \mathcal{V}$.

1: **for** $t = 1$ to $T - 1$ **do** in parallel for each node $i \in \mathcal{V}$

2:     Compress the difference vector $q_i^t = Q(x_i^t - \hat{x}_i^t)$ and update $\hat{x}_i^{t+1} = \hat{x}_i^t + q_i^t$.

3:     Spread $q_i^t$ and receive $q_j^t, j \in \mathcal{N}_i$. Update $s_i^{t+1} = s_i^t + \sum_{j \in \mathcal{N}_i} a_{ij} q_j^t$.

4:     Receive the one-point bandit feedback and construct $g_i^t = \frac{d}{\epsilon} f_i^t(x_i^t + \epsilon u_i^t) u_i^t$.

5:     Update the decision variable $x_i^{t+1} = P_{(1-\varsigma)\mathcal{K}} \left( x_i^t + \gamma(s_i^{t+1} - \hat{x}_i^{t+1}) - \eta_t g_i^t \right)$.

**Output:** $\{x_i^t\}_{t=1}^T$

---

**Assumption 7.** *For each $i \in \mathcal{V}$ and $t = 1, 2, ..., T$, the loss function $f_i^t$ is convex and $l$-Lipschitz continuous in $\mathcal{K}$, i.e., $|f_i^t(x) - f_i^t(y)| \le l\|x - y\|$, $\forall x, y \in \mathcal{K}$.*

Assumptions 6 and 7 lead to an uniform upper bound on the function value, i.e., there exists a constant $B > 0$ such that $\max_{x,i,t} |f_i^t(x)| \le B$. Then we establish the expected regret bounds of Algorithm 2 for convex and strongly convex losses, respectively.

**Theorem 2.** *Let common Assumptions 1, 2, 6 and 7 hold. Consider Algorithm 2 with the consensus stepsize $\gamma$ chosen in (5). Denote*

$$H = 4\sqrt{3} \left( \sqrt{N} + 2\sqrt{3}\gamma^{-1}\delta^{-1} + 1 \right) \left( 1 + \gamma^{-1}\delta^{-1} + \omega^{-1} \right). \tag{8}$$

*(i) (Convex case) Take the gradient descent stepsize $\eta_t = \frac{2R\epsilon}{dB\sqrt{t+c}}$ for a constant $c \ge \frac{8}{3\gamma\delta}$, $\epsilon = \left( \frac{(1+4H)dBR}{2\left(l + \frac{B}{r}\right)} \right)^{\frac{1}{2}} \frac{(T+c)^{\frac{1}{4}}}{T^{\frac{1}{2}}}$ and $\zeta = \frac{\epsilon}{r}$. Then for each $j \in \mathcal{V}$ and $T \ge 1$,*

$$\mathbb{E}\left[ R(j, T) \right] \le 2NT^{\frac{1}{2}}(T + c)^{\frac{1}{4}} \sqrt{2(1 + 4H)(l + B/r)\, dBR}. \tag{9}$$

*(ii) (Strongly convex case) With additional Assumption 5, take the gradient descent stepsize $\eta_t = \frac{1}{\mu(t+c)}$ for a constant $c \ge \frac{16}{3\gamma\delta}$, $\epsilon = \left( \frac{Hd^2B^2 \ln(T+c)}{(l + \frac{B}{r})\mu T} \right)^{\frac{1}{3}}$ and $\zeta = \frac{\epsilon}{r}$. Then for each $j \in \mathcal{V}$ and $T \ge 1$,*

$$\mathbb{E}\left[ R(j, T) \right] \le 4\mu c R^2 + 3N \left( Hd^2B^2\mu^{-1} \right)^{\frac{1}{3}} (l + B/r)^{\frac{2}{3}} T^{\frac{2}{3}} \ln^{\frac{1}{3}}(T + c). \tag{10}$$

Theorem 2 shows that Algorithm 2 is also no-regret, and it achieves the regret bounds $\mathcal{O}(d^{1/2}N^{5/4}T^{3/4})$ and $\mathcal{O}(d^{2/3}N^{7/6}T^{2/3}\ln^{1/3}(T))$ for convex losses and strongly convex losses, respectively, which match the bounds obtained by [11]. The regrets are scaled with $\left( \omega^{-2}N^{1/2} + \omega^{-4} \right)^{1/2}$ and $\left( \omega^{-2}N^{1/2} + \omega^{-4} \right)^{1/3}$ for convex and strongly convex losses, which indicates that the influence of the compression ratio $\omega$ on the regret bounds in the one-point bandit setting is less than that in the full information setting.

## 4 Two-point Bandit Feedback

In the two-point bandit feedback, agent $i$ can query the loss function values at two points around $x_i^t$. Like the procedure in [23], let agent $i$ pick a unit-norm vector $u_i^t \in \mathbb{R}^d$ uniformly at random, query the values of $f_i^t$ at $y_{i,1}^t = x_i^t + \epsilon u_i^t$ and $y_{i,2}^t = x_i^t - \epsilon u_i^t$, and estimate the gradient as $g_i^t = \frac{d}{2\epsilon} \left( f_i^t(y_{i,1}^t) - f_i^t(y_{i,2}^t) \right) u_i^t$. Then we obtain DC-DO2BD as a variant of DC-DOBD.

In the two-point bandit setting, the regret of node $j \in \mathcal{V}$ is modified as

$$R_2(j, T) = \sum_{t=1}^T \sum_{i=1}^N \frac{f_i^t(y_{j,1}^t) + f_i^t(y_{j,2}^t)}{2} - \sum_{t=1}^T \sum_{i=1}^N f_i^t(x^*). \tag{11}$$

---

**Algorithm 3** Distributed Online Two-point Bandit Gradient Descent with Difference Compression (DC-DO2BD)

---

**Input:** consensus stepsize $\gamma$, GD stepsizes $\{\eta_t\}_{t=1}^T$, time $T$, exploration parameter $\epsilon$, shrinkage parameter $\zeta$
**Initialize:** set $x_i^1 = \mathbf{0}, \hat{x}_i^1 = \mathbf{0}, s_i^1 = \mathbf{0}$, for each node $i \in \mathcal{V}$.
1: **for** $t = 1$ to $T - 1$ **do** in parallel for each node $i \in \mathcal{V}$
2:      Compress the difference vector $q_i^t = Q(x_i^t - \hat{x}_i^t)$ and update $\hat{x}_i^{t+1} = \hat{x}_i^t + q_i^t$.
3:      Spread $q_i^t$ and receive $q_j^t, j \in \mathcal{N}_i$. Update $s_i^{t+1} = s_i^t + \sum_{j \in \mathcal{N}_i} a_{ij} q_j^t$.
4:      Receive the two-point feedback and construct $g_i^t = \frac{d}{2\epsilon} \left( f_i^t(x_i^t + \epsilon u_i^t) - f_i^t(x_i^t - \epsilon u_i^t) \right) u_i^t$.
5:      Update the decision variable $x_i^{t+1} = P_{(1-\zeta)\mathcal{K}} \left( x_i^t + \gamma(s_i^{t+1} - \hat{x}_i^{t+1}) - \eta_t g_i^t \right)$.
**Output:** $\{x_i^t\}_{t=1}^T$

---

**Theorem 3.** *Let common Assumptions 1, 2, 6 and 7 hold. Consider Algorithm 3 with the consensus stepsize $\gamma$ chosen in (5) and $H$ defined in (8).*

*(i) (Convex case) Take the gradient descent stepsize $\eta_t = \frac{2R}{dl\sqrt{t+c}}$ for a constant $c \geq \frac{8}{3\gamma\delta}$, $\epsilon = \frac{1}{\sqrt{T}}$ and $\zeta = \frac{\epsilon}{r}$. Then for each $j \in \mathcal{V}$ and $T \geq 1$,*

$$\mathbb{E}\left[\mathrm{R}_2(j,T)\right] \leq (1 + 4H)RNdl\sqrt{T+c} + (3 + 2R/r)\,Ndl\sqrt{T}. \tag{12}$$

*(ii) (Strongly convex case) With additional Assumption 5, take the gradient descent stepsize $\eta_t = \frac{1}{\mu(t+c)}$ for a constant $c \geq \frac{16}{3\gamma\delta}$, $\epsilon = \frac{\ln(T)}{T}$ and $\zeta = \frac{\epsilon}{r}$. Then for each $j \in \mathcal{V}$ and $T \geq 1$,*

$$\mathbb{E}\left[\mathrm{R}_2(j,T)\right] \leq 4\mu c R^2 + \mu^{-1} N d^2 l^2 H \ln(T+c) + (3 + 2R/r)\,Ndl\ln(T). \tag{13}$$

Theorem 3 shows that Algorithm 3 achieves $\mathcal{O}((\omega^{-2}N^{1/2} + \omega^{-4})Nd\sqrt{T})$ and $\mathcal{O}((\omega^{-2}N^{1/2} + \omega^{-4})Nd^2\ln(T))$ regret bounds for convex and strongly convex losses, respectively, which recovers the regret bounds $\mathcal{O}(\sqrt{T})$ (convex) and $\mathcal{O}(\ln(T))$ (strongly convex) in the full information case, while the constants are larger than those of Theorem 1.

## 5 Numerical Experiments

In this section, we evaluate the three proposed algorithms on a real-world online problem. A prominent example is the diabetes prediction task, which aims to diagnose diabetes through several risk factors. Consider the distributed online regularized logistic regression with the local loss function

$$f_i^t(x) = \sum_{j=1}^S \log\left(1 + \exp\left(-b_{i,j}^t \left\langle a_{i,j}^t, x \right\rangle\right)\right) + \frac{\mu}{2}\|x\|^2, \tag{14}$$

where $\mu$ is the regularization parameter, and a batch data samples $\{(a_{i,j}, b_{i,j})\}_{j=1}^S$ are revealed to agent $i$ at time $t$. We adopt *diabetes-binary-BRFSS2015* dataset with 70692 instances, 21 features, and 2 labels from Kaggle.[2] Here, $a_{i,j} \in \mathbb{R}^d$ with $d = 21$, and $b_{i,j} \in \{-1, 1\}$. We standardize the data samples and distribute them evenly among $N$ agents under the sorted setting, i.e., each agent only gets data samples from one class. The connected communication network $\mathcal{G}(N, M)$ with $N$ nodes and $M$ edges is generated randomly by tool *NetworkX* [42], and then we use the Metropolis rule [43] to construct the connectivity matrix $A$ to satisfy Assumption 1. We repeat each experiment ten times and depict the mean curve.[3] The parameters selection details are given in Appendix E.

**Comparison experiment** We run our algorithms DC-DOGD, DC-DOBD, DC-DO2BD, and make comparisons with ECD-AMSGrad [20], for the convex case ($\mu = 0$) and strongly convex case ($\mu = 1$). The compressor type, the compression ratio, and the communication network are kept the same. Take the setting of compressor $\mathrm{QSGD}_2$ with $\omega = 0.3$ over $\mathcal{G}(9, 18)$ as an example. We plot the time averaged maximum regret $SR(T) := {}^{max_j \mathrm{R}(j,T)}/_T$ versus the time horizon $T$ and versus

---

[2]The data set is from https://www.kaggle.com/code/encode0/diabetes-prediction-and-risk-factors-evaluation.
[3]All experiments are performed on a 64-bit Windows platform with the Intel(R) Core(TM) i7-6850K 3.6Ghz CPU. The codes are provided in the supplementary materials.

the total number of transmitted bits in Fig. 1, where the best solution in the hindsight $x^*$ is obtained by *LogisticRegression* optimizer from scikit-learn [44].

Fig. 1 shows that the time averaged regrets of DC-DOGD, DC-DOBD, and DC-DO2BD go to zero as $T$ goes to infinity, which is in agreement with the theoretical results that our algorithms are no-regret. Among the three proposed algorithms, the one-bandit feedback has the worst performance, while using two-bandit information can improve the performance and even reach that of the full information feedback. ECD-AMSGrad gets deteriorated in the first few steps because the inverse of the second raw moment estimation is large and this algorithm does not have a projection to restrict variables. Then, ECD-AMSGrad declines fast, while its time-average regret can not reach zero. Clearly, DC-DOGD and DC-DO2BD significantly outperform ECD-AMSGrad.

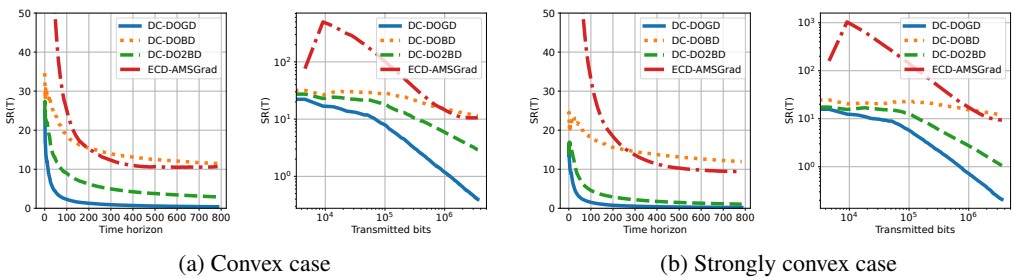

(a) Convex case           (b) Strongly convex case

Figure 1: Comparison of algorithms DC-DOGD, DC-DOBD, DC-DO2BD, and ECD-AMSGrad with $\text{QSGD}_2$, $\omega = 0.3$, $\mathcal{G}(9, 18)$.

**Impacts of compression ratio and compressor type**   Fixing the compressor type ($\text{Top}_k$) and the graph $\mathcal{G}(9, 18)$, we run DC-DOGD with different compression ratios ($\omega = 0.05, 0.1, 0.5$) for strongly convex losses.[4] We consider DAOL [8] as the baseline, which is with exact communication and is the special case of DC-DOGD with $\omega = 1$. Fig. 2a shows that the greater the compression degree is (less $\omega$), the more iteration rounds are needed to reach a certain average regret, while the total transmitted bits are actually the fewer. DC-DOGD with $\omega = 0.05$ have approximately $8\times$ reduction on transmitted bits to reach a certain average regret compared with DAOL.

Then, we fix the compression ratio ($\omega = 0.3$) and the graph $\mathcal{G}(9, 18)$, and run DC-DOGD with different compressors ($\text{Top}_k$, $\text{Rand}_k$, $\text{RGossip}_p$, $\text{GSGD}_s$) for strongly convex losses. Fig. 2b shows that $\text{Rand}_k$ and $\text{RGossip}_p$ have almost the same performance. It is expected, since $\text{Rand}_k$ allows each element of the vector to has the probability $\omega = k/d$ to be chosen to be transmitted, which is equivalent to randomly transmitting the whole vector with the probability $p = \omega$. Besides, Figs. 2b shows that $\text{Top}_k$ has better performance than $\text{Rand}_k$, which is in line with the intuition that the largest $k$ coordinates contain more useful information than arbitrary $k$ coordinates. In addition, quantization $\text{GSGD}_s$ performs better in reducing the total transmitted bits than sparsification under the same compression ratio.

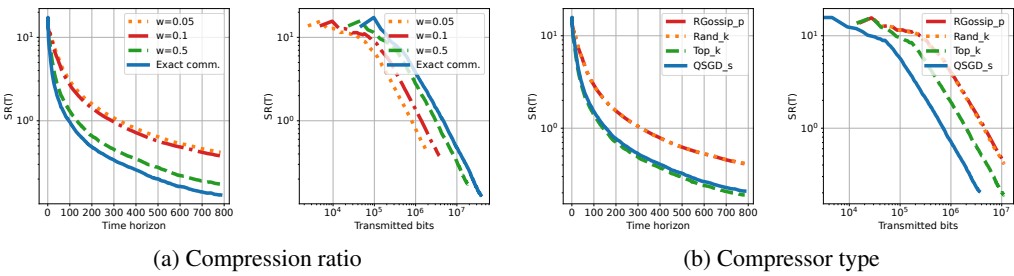

(a) Compression ratio           (b) Compressor type

Figure 2: The impacts of compression ratio and compressor type for DC-DOGD over $\mathcal{G}(9, 18)$ in the strongly convex case.

---

[4]Since the trajectories in the convex case and strongly convex case share similar trends, here we only present the experiment results in the strongly convex case, for space limitation.

**Impacts of topology and node number**    The communication network topology affects the parameters $\delta$ and $\beta$, which affect the choice of the stepsizes $\eta_t$ and $\gamma$ as well as the algorithm performance. We take DC-DOGD in the strongly convex case as an example. Fixing the compressor ($\text{Top}_1$) and the compression ratio ($\omega = 0.05$), we assess DC-DOGD over three basic topologies, namely ring, $\mathcal{G}(N, 2N)$, and full connected. Fig. 3a shows that full connected topology exhibits the best regret with respect to the time horizon, while $\mathcal{G}(N, 2N)$ performs slightly better in the sense of tranmitted bits. A sparser topology with fewer edges uses fewer bits to transmit information in each iteration, while more iteration rounds are needed for decision consensus. Thus, there will be a tradeoff case by case.

Finally, we let the node number $N$ vary from to 10 to 50, and run DC-DOGD, DC-DOBD, DC-DO2BD with the same compressor $\text{Top}_2$, the same compression ratio $\omega = 0.1$, over the full connected graph, for convex losses and strongly convex losses. We plot the node averaged regret $AR(T) := {}^{max_j \, \text{R}(j,T)}\!/_N$ versus the node number $N$ in Fig. 3b.

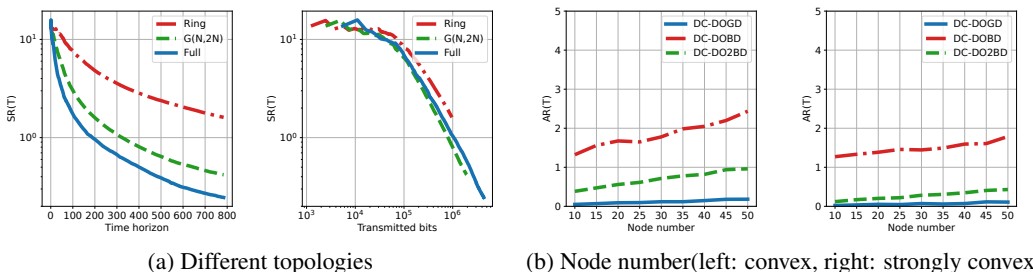

(a) Different topologies                (b) Node number(left: convex, right: strongly convex)

Figure 3: The impacts of topology and node number.

# 6   Conclusions

In this paper, we considered DOCO with the full information feedback, one-point and two-points bandit feedback. We designed provably no-regret distributed online algorithms that work with $\omega$-contracted compressors. The obtained regret bounds for both convex and strongly convex losses matched those of uncompressed algorithms in the literature. We further assessed the influence of the compressor type and the compression ratio $\omega$ on the regrets, and showed that $\omega$ can be used to balance the iteration rounds and the transmitted bits according to the bandwidth.

A limitation of this work is that the obtained regret bounds show high order inverse dependence on the compression ratio, which are pretty conservative and may be further improved. Besides, the impact of topology sparsity is worth further investigation. We believe this paper is an important step in this direction. Future research includes extensions from fixed graphs to time-varying graphs and combinations with other compression schemes such as extrapolation compression.

## Acknowledgments and Disclosure of Funding

This work is supported in part by Shanghai Municipal Science and Technology Major Project under Grant 2021SHZDZX0100 and the National Natural Science Foundation of China under Grant 62173250, and in part by Australian Research Council under Grant DP190103615 and LP210200473.

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
