# A Basic Inequalities

Firstly, we present some preliminary inequalities that will be frequently used in the subsequent proofs.

**Fact 1** (Cauchy-Schwarz Inequality)**.** *For any $x, y \in \mathbb{R}^d$,*

$$| \langle x, y \rangle | \leq \|x\| \cdot \|y\|. \tag{15}$$

**Fact 2.** *For arbitrary set of $N$ vectors $\{x_i\}_{i=1}^N, x_i \in \mathbb{R}^d$,*

$$\left\| \sum_{i=1}^N x_i \right\|^2 \leq N \sum_{i=1}^N \|x_i\|^2. \tag{16}$$

**Fact 3.** *For any $x, y \in \mathbb{R}^d$,*

$$\|x + y\|^2 \leq (1 + \alpha)\|x\|^2 + (1 + \alpha^{-1})\|x\|^2, \quad \forall \alpha > 0. \tag{17}$$

**Fact 4.** *Given a convex set $\mathcal{K} \in \mathbb{R}^d$, the projection operator satisfies the following properties*

$$\text{(i)} \quad \|P_\mathcal{K}(x) - P_\mathcal{K}(y)\| \leq \|x - y\|, \qquad \forall x, y \in \mathbb{R}^d. \tag{18}$$

$$\text{(ii)} \quad \|P_\mathcal{K}(x) - x\| \leq \|x - y\|, \qquad \forall x \in \mathbb{R}^d, y \in \mathcal{K}. \tag{19}$$

$$\text{(iii)} \quad \langle P_\mathcal{K}(x) - x, x - y \rangle \leq -\|P_\mathcal{K}(x) - x\|^2 \leq 0, \qquad \forall x \in \mathbb{R}^d, y \in \mathcal{K}. \tag{20}$$

**Fact 5** (Jensen's Inequality)**.** *Given a convex function $f$ and a random variable $x$, then*

$$f(\mathbb{E}[x]) \leq \mathbb{E}[f(x)]. \tag{21}$$

# B Proofs of Section 2

Define

$$\tilde{x}_i^{t+1} := x_i^t + \gamma \sum_{j \in \mathcal{N}_i} a_{ij}(\hat{x}_j^{t+1} - \hat{x}_i^{t+1}) - \eta_t \nabla f_i^t(x_i^t), \tag{22}$$

$$r_i^{t+1} := P_\mathcal{K}\left(\tilde{x}_i^{t+1}\right) - \tilde{x}_i^{t+1}, \tag{23}$$

$$\bar{x}^t := \frac{1}{N} \sum_{i=1}^N x_i^t, \tag{24}$$

and then

$$x_i^{t+1} = P_\mathcal{K}\left(\tilde{x}_i^{t+1}\right) = \tilde{x}_i^{t+1} + r_i^{t+1}. \tag{25}$$

For notational simplicity, define matrices

$$X^t := \text{col}\{x_1^t, \cdots, x_N^t\}, \quad \widetilde{X}^t := \text{col}\{\tilde{x}_1^t, \cdots, \tilde{x}_N^t\}, \quad \bar{X}^t := \text{col}\{\bar{x}^t, \cdots, \bar{x}^t\},$$
$$R^t := \text{col}\{r_1^t, \cdots, r_N^t\}, \quad \nabla F^t(X^t) := \text{col}\{\nabla f_1^t(x_1^t), \cdots, \nabla f_N^t(x_N^t)\}.$$

Denote by $1_N$ the $N$-dimension column vector with all components being one, and $M := \frac{1}{N} 1_N 1_N^\top, \boldsymbol{M} := M \otimes I_d$. Then $\bar{X}^t = \boldsymbol{M} X^t$. Define the Laplacian matrix $L := I_N - A$ and $\boldsymbol{L} := L \otimes I_d, \boldsymbol{I} := I_N \otimes I_d$, where $\otimes$ is the Kronecher product. Denote by $\boldsymbol{L}_i$ the $i$-th row of $\boldsymbol{L}$. Then by Remark 1, Algorithm 1 can be written in the matrix form as

$$\hat{X}^{t+1} = \hat{X}^t + Q(X^t - \hat{X}^t), \tag{26}$$

$$X^{t+1} = P_\mathcal{K}\left(X^t - \gamma \boldsymbol{L}\hat{X}^{t+1} - \eta_t \nabla F^t(X^t)\right) \tag{27}$$

$$= \widetilde{X}^{t+1} + R^{t+1}. \tag{28}$$

To begin with, we consider general regret bounds.

**Lemma 1.** *Consider Algorithm 1 with non-increasing gradient descent stepsizes $\{\eta_t\}_{t=1}^T$.*

*(i) (Convex case) Suppose Assumptions 1, 3, 4 hold. Then for each $j \in \mathcal{V}$:*

$$\mathrm{R}(j,T) \le \frac{ND^2}{2\eta_T} + NG^2 \sum_{t=1}^T \eta_t + (2\sqrt{N}+N)G \sum_{t=1}^T \|X^t - \bar{X}^t\|$$
$$+ \sum_{t=1}^T \frac{1}{2\eta_t}\left(\left\|\bar{X}^t - \tilde{X}^{t+1}\right\|^2 + 3\left\|R^{t+1}\right\|^2\right). \tag{29}$$

*(ii) (Strongly convex case) Suppose Assumptions 1, 3, 4, 5 hold and $\eta_t = \frac{1}{\mu(t+c)}$ for a constant $c \ge 0$. Then for each $j \in \mathcal{V}$:*

$$\mathrm{R}(j,T) \le \mu c D^2 + NG^2 \sum_{t=1}^T \eta_t + (2\sqrt{N}+N)G \sum_{t=1}^T \|X^t - \bar{X}^t\|$$
$$+ \sum_{t=1}^T \frac{1}{2\eta_t}\left(\left\|\bar{X}^t - \tilde{X}^{t+1}\right\|^2 + 3\left\|R^{t+1}\right\|^2\right). \tag{30}$$

*Proof.* Because $\sum_{i=1}^N \sum_{j \in \mathcal{N}_i} a_{ij}(\hat{x}_j^{t+1} - \hat{x}_i^{t+1}) = 0$ under Assumption 1, with the introduction of the projection error $r_i^{t+1}$, we can write

$$\bar{x}^{t+1} = \frac{1}{N}\sum_{i=1}^N \left(\tilde{x}_i^{t+1} + r_i^{t+1}\right) = \bar{x}^t - \frac{\eta_t}{N}\sum_{i=1}^N \nabla f_i^t(x_i^t) + \frac{1}{N}\sum_{i=1}^N r_i^{t+1}.$$

Denote by $x^*$ the best decision in the hindsight, i.e., $x^* = \arg\min_{x \in \mathcal{K}} \sum_{t=1}^T \sum_{i=1}^N f_i^t(x)$. Then,

$$\|\bar{x}^{t+1} - x^*\|^2 = \|\bar{x}^t - x^*\|^2 + \frac{1}{N^2}\left\|\sum_{i=1}^N r_i^{t+1} - \eta_t \sum_{i=1}^N \nabla f_i^t(x_i^t)\right\|^2$$
$$+ \frac{2}{N}\sum_{i=1}^N \left\langle r_i^{t+1}, \bar{x}^t - x^*\right\rangle - \frac{2\eta_t}{N}\sum_{i=1}^N \left\langle \nabla f_i^t(x_i^t), \bar{x}^t - x^*\right\rangle. \tag{31}$$

Under Assumption 4, we estimate the second term

$$\frac{1}{N^2}\left\|\sum_{i=1}^N r_i^{t+1} - \eta_t \sum_{i=1}^N \nabla f_i^t(x_i^t)\right\|^2 = \frac{1}{N^2}\left(2\left\|\sum_{i=1}^N r_i^{t+1}\right\|^2 + 2\eta_t^2 \left\|\sum_{i=1}^N \nabla f_i^t(x_i^t)\right\|^2\right)$$
$$\overset{(16)}{\le} \frac{1}{N^2}\left(2N\sum_{i=1}^N \left\|r_i^{t+1}\right\|^2 + 2\eta_t^2 N^2 G^2\right) = \frac{2}{N}\left\|R^{t+1}\right\|^2 + 2\eta_t^2 G^2. \tag{32}$$

Then we come to the third term. Noting that $x^* \in \mathcal{K}$, by using the definition of $r_i^{t+1}$ and the projection property (iii), we have

$$\sum_{i=1}^N \left\langle r_i^{t+1}, \bar{x}^t - x^*\right\rangle = \sum_{i=1}^N \left(\left\langle r_i^{t+1}, \bar{x}^t - \tilde{x}_i^{t+1}\right\rangle + \left\langle P_{\mathcal{K}}\left(\tilde{x}_i^{t+1}\right) - \tilde{x}_i^{t+1}, \tilde{x}_i^{t+1} - x^*\right\rangle\right) \tag{33}$$

$$\overset{(20)}{\le} \sum_{i=1}^N \left\langle r_i^{t+1}, \bar{x}^t - \tilde{x}_i^{t+1}\right\rangle \le \sum_{i=1}^N \frac{1}{2}\left(\left\|r_i^{t+1}\right\|^2 + \left\|\bar{x}^t - \tilde{x}_i^{t+1}\right\|^2\right) = \frac{1}{2}\left(\left\|R^{t+1}\right\|^2 + \left\|\bar{X}^t - \tilde{X}^{t+1}\right\|^2\right).$$

Next we turn to the fourth term. Under Assumption 4,

$$f_i^t(x_i^t) \ge f_i^t(x_j^t) + \left\langle \nabla f_i^t(x_j^t), x_i^t - x_j^t\right\rangle \ge f_i^t(x_j^t) - G\|x_i^t - x_j^t\|,$$

and hence,

$$-\left\langle \nabla f_i^t(x_i^t), \bar{x}^t - x^*\right\rangle = \left\langle \nabla f_i^t(x_i^t), x^* - x_i^t\right\rangle + \left\langle \nabla f_i^t(x_i^t), x_i^t - \bar{x}^t\right\rangle$$
$$\le f_i^t(x^*) - f_i^t(x_i^t) - \frac{\mu}{2}\|x^* - x_i^t\|^2 + G\|x_i^t - \bar{x}^t\|$$
$$\le f_i^t(x^*) - f_i^t(x_j^t) + G\|x_i^t - x_j^t\| - \frac{\mu}{2}\|x^* - x_i^t\|^2 + G\|x_i^t - \bar{x}^t\|, \tag{34}$$

where $\mu > 0$ for the strongly convex case and $\mu \equiv 0$ for the convex case. Summing up (34) over $i = 1, \cdots, N$ with the fact that

$$\sum_{i=1}^{N} \|x_i^t - x_j^t\| \leq \sum_{i=1}^{N} \|x_i^t - \bar{x}^t\| + N\|\bar{x}^t - x_j^t\| \leq \sqrt{N}\|X^t - \bar{X}^t\| + N\|X^t - \bar{X}^t\|,$$

$$\sum_{i=1}^{N} \|x^* - x_i^t\|^2 \geq \frac{1}{N} \left\|\sum_{i=1}^{N}(x^* - x_i^t)\right\|^2 \geq \frac{1}{N} \|Nx^* - N\bar{x}^t\|^2 = N \|x^* - \bar{x}^t\|^2,$$

we have

$$-\sum_{i=1}^{N} \langle \nabla f_i^t(x_i^t), \bar{x}^t - x^* \rangle \leq \sum_{i=1}^{N} \left(f_i^t(x^*) - f_i^t(x_j^t)\right) + (2\sqrt{N}+N)G\|X^t - \bar{X}^t\| - \frac{N\mu}{2} \|x^* - \bar{x}^t\|^2.$$
(35)

By substituting (32), (33), and (35) into (31), we derive

$$\|\bar{x}^{t+1} - x^*\|^2 \leq \|\bar{x}^t - x^*\|^2 + \frac{2}{N} \|R^{t+1}\|^2 + 2\eta_t^2 G^2 + \frac{1}{N}\left(\|R^{t+1}\|^2 + \left\|\bar{X}^t - \tilde{X}^{t+1}\right\|^2\right)$$

$$+ \frac{2\eta_t}{N}\left(\sum_{i=1}^{N}\left(f_i^t(x^*) - f_i^t(x_j^t)\right) + (2\sqrt{N}+N)G\|X^t - \bar{X}^t\| - \frac{N\mu}{2}\|x^* - \bar{x}^t\|^2\right).$$

By rearranging the terms,

$$\sum_{i=1}^{N}\left(f_i^t(x_j^t) - f_i^t(x^*)\right) \leq \frac{N}{2}\left(\left(\frac{1}{\eta_t} - \mu\right)\|\bar{x}^t - x^*\|^2 - \frac{1}{\eta_t}\|\bar{x}^{t+1} - x^*\|^2\right)$$

$$+ NG^2\eta_t + (2\sqrt{N}+N)G\|X^t - \bar{X}^t\| + \frac{1}{2\eta_t}\left(\left\|\bar{X}^t - \tilde{X}^{t+1}\right\|^2 + 3\left\|R^{t+1}\right\|^2\right).$$

Summing up the above inequality over $t = 1, \cdots, T$ gives

$$\sum_{t=1}^{T}\sum_{i=1}^{N}\left(f_i^t(x_j^t) - f_i^t(x^*)\right) \leq \frac{N}{2}\sum_{t=1}^{T}\left(\frac{1}{\eta_t} - \frac{1}{\eta_{t-1}} - \mu\right)\|\bar{x}^t - x^*\|^2 + NG^2\sum_{t=1}^{T}\eta_t$$

$$+ (2\sqrt{N}+N)G\sum_{t=1}^{T}\|X^t - \bar{X}^t\| + \sum_{t=1}^{T}\frac{1}{2\eta_t}\left(\left\|\bar{X}^t - \tilde{X}^{t+1}\right\|^2 + 3\left\|R^{t+1}\right\|^2\right), \frac{1}{\eta_0} \triangleq 0$$
(36)

(i) In the convex case, $\mu \equiv 0$. Using Assumption 3 with the non-increasing of $\{\eta_t\}_{t=1}^{T}$, we have

$$\sum_{t=1}^{T}\left(\frac{1}{\eta_t} - \frac{1}{\eta_{t-1}}\right)\|\bar{x}^t - x^*\|^2 \leq \sum_{t=1}^{T}\left(\frac{1}{\eta_t} - \frac{1}{\eta_{t-1}}\right)D^2 \leq \frac{D^2}{\eta_T}.$$
(37)

By substituting (37) into (36), we derive (29).

(ii) Under Assumption 5, $\mu > 0$, and thus, $\eta_t = \frac{1}{\mu(t+c)}$ implies $\frac{1}{\eta_t} - \frac{1}{\eta_{t-1}} - \mu = 0, \forall t \geq 2$. Then

$$\sum_{t=1}^{T}\left(\frac{1}{\eta_t} - \frac{1}{\eta_{t-1}} - \mu\right)\|\bar{x}^t - x^*\|^2 = \left(\frac{1}{\eta_1} - \mu\right)\|\bar{x}^1 - x^*\|^2 \leq \mu c D^2.$$
(38)

By substituting (38) into (36), we derive (30).

$\square$

The following key lemma analyzes the relationship between the projection error, the consensus error, and the compressor error, and makes it possible to control these errors by the consensus stepsize $\gamma$ and the gradient descent stepsize $\eta_t$.

**Lemma 2.** *Suppose Assumptions 1, 2 and 4 hold. Consider Algorithm 1 with the consensus stepsize* $\gamma \in (0,1]$ *and arbitrary gradient descent stepsizes* $\{\eta_t\}_{t=1}^{T}$.

(i) $\mathbb{E}_Q \left\| R^{t+1} \right\|^2 \leq 2(1-\omega)\beta^2\gamma^2 \left\| X^t - \hat{X}^t \right\|^2 + 2NG^2\eta_t^2.$ \hfill (39)

(ii) $\mathbb{E}_Q \left\| X^{t+1} - \bar{X}^{t+1} \right\|^2 \leq (1-\gamma\delta)\mathbb{E}_Q \left\| X^t - \bar{X}^t \right\|^2 + 9\left(1 + \frac{2}{\gamma\delta}\right) NG^2\eta_t^2$

$$+ 9\left(1 + \frac{2}{\delta}\right)(1-\omega)\beta^2\gamma \left\| X^t - \hat{X}^t \right\|^2.$$ \hfill (40)

(iii) $\mathbb{E}_Q \left\| X^{t+1} - \hat{X}^{t+1} \right\|^2 \leq 3\left(1 + \frac{2}{\omega}\right)\gamma\beta^2 \mathbb{E}_Q \left\| X^t - \bar{X}^t \right\|^2 + 9\left(1 + \frac{2}{\omega}\right)NG^2\eta_t^2$

$$+ \left(\left(1 + \frac{\omega}{2}\right)(1-\omega)(1 + (\beta^2 + 2\beta)\gamma) + 6\left(1 + \frac{2}{\omega}\right)(1-\omega)\beta^2\gamma\right)\left\| X^t - \hat{X}^t \right\|^2.$$ \hfill (41)

*Proof.* First of all, by Assumption 2 and the update rule of $\hat{X}^{t+1}$,

$$\mathbb{E}_Q \left\| X^t - \hat{X}^{t+1} \right\|^2 \stackrel{(26)}{=} \mathbb{E}_Q \left\| X^t - \hat{X}^t - Q(X^t - \hat{X}^t) \right\|^2 \leq (1-\omega)\left\| X^t - \hat{X}^t \right\|^2.$$ \hfill (42)

(i) By Assumption 1, $\sum_{j\in\mathcal{N}_i} a_{ij} = 1$. Since $\mathcal{K}$ is a convex set and $x_j^t \in \mathcal{K}$, we have $\sum_{j\in\mathcal{N}_i} a_{ij}x_j^t \in \mathcal{K}$ and $(1-\gamma)x_i^t + \gamma\sum_{j\in\mathcal{N}_i} a_{ij}x_j^t \in \mathcal{K}$ for $\gamma \in (0,1]$. By the projection property (ii),

$$\left\| r_i^{t+1} \right\| = \left\| P_\mathcal{K}\left(\tilde{x}_i^{t+1}\right) - \tilde{x}_i^{t+1} \right\|$$

$$\stackrel{(19)}{\leq} \left\| (1-\gamma)x_i^t + \gamma\sum_{j\in\mathcal{N}_i} a_{ij}x_j^t - \tilde{x}_i^{t+1} \right\|$$

$$= \left\| x_i^t + \gamma\sum_{j\in\mathcal{N}_i} a_{ij}(x_j^t - x_i^t) - \left( x_i^t + \gamma\sum_{j\in\mathcal{N}_i} a_{ij}(\hat{x}_j^{t+1} - \hat{x}_i^{t+1}) - \eta_t \nabla f_i^t(x_i^t) \right) \right\|$$

$$= \left\| -\gamma \boldsymbol{L}_i\left(X^t - \hat{X}^t\right) + \eta_t \nabla f_i^t(x_i^t) \right\|.$$

Then, we can estimate the total projection error as

$$\left\| R^{t+1} \right\|^2 = \sum_{i=1}^{N} \left\| r_i^{t+1} \right\|^2 \leq \sum_{i=1}^{N} \left( 2\left\| \gamma \boldsymbol{L}_i\left(X^t - \hat{X}^{t+1}\right) \right\|^2 + 2\left\| \eta_t \nabla f_i^t(x_i^t) \right\|^2 \right)$$

$$\leq 2\left\| \gamma \boldsymbol{L}\left(X^t - \hat{X}^{t+1}\right) \right\|^2 + 2NG^2\eta_t^2 = 2\gamma^2\beta^2 \left\| X^t - \hat{X}^{t+1} \right\|^2 + 2NG^2\eta_t^2,$$ \hfill (43)

which is controlled by $\gamma$ and $\eta_t$. By taking expectation over the internal randomness of the compressor $Q$ with respect to the above inequality and using (42), we derive (39).

(ii) Under Assumption 1, $\boldsymbol{ML} = \boldsymbol{LM} = 0$ and $\boldsymbol{L}\bar{X}^t = \boldsymbol{LM}X^t = 0$. By the update rule of $X^{t+1}$,

$$\left\| X^{t+1} - \bar{X}^{t+1} \right\|^2$$

$$\stackrel{(28)}{=} \left\| X^t - \gamma\boldsymbol{L}\hat{X}^{t+1} - \eta_t\nabla F^t(X^t) + R^{t+1} - \boldsymbol{M}\left( X^t - \gamma\boldsymbol{L}\hat{X}^{t+1} - \eta_t\nabla F^t(X^t) + R^{t+1} \right) \right\|^2$$

$$= \left\| X^t - \bar{X}^t - \gamma\boldsymbol{L}\hat{X}^{t+1} - \eta_t(\boldsymbol{I} - \boldsymbol{M})\nabla F^t(X^t) + (\boldsymbol{I} - \boldsymbol{M})R^{t+1} \right\|^2$$

$$= \left\| (\boldsymbol{I} - \gamma\boldsymbol{L})\left(X^t - \bar{X}^t\right) - \gamma\boldsymbol{L}\left(\hat{X}^{t+1} - X^t\right) - \eta_t(\boldsymbol{I} - \boldsymbol{M})\nabla F^t(X^t) + (\boldsymbol{I} - \boldsymbol{M})R^{t+1} \right\|^2$$

$$\stackrel{(17)}{\leq} \left(1 + \frac{\gamma\delta}{2}\right)\left\| (\boldsymbol{I} - \gamma\boldsymbol{L})\left(X^t - \bar{X}^t\right) \right\|^2$$

$$+ \left(1 + \frac{2}{\gamma\delta}\right)\left\| -\gamma\boldsymbol{L}\left(\hat{X}^{t+1} - X^t\right) - \eta_t(\boldsymbol{I} - \boldsymbol{M})\nabla F^t(X^t) + (\boldsymbol{I} - \boldsymbol{M})R^{t+1} \right\|^2.$$ \hfill (44)

The first term can be estimated by

$$\left\| (\boldsymbol{I} - \gamma \boldsymbol{L}) \left( X^t - \bar{X}^t \right) \right\| = \left\| ((1 - \gamma)\boldsymbol{I} + \gamma \boldsymbol{A}) \left( X^t - \bar{X}^t \right) \right\|$$
$$= (1 - \gamma) \left\| X^t - \bar{X}^t \right\| + \gamma \left\| (\boldsymbol{A} - \boldsymbol{M}) \left( X^t - \bar{X}^t \right) \right\|$$
$$\leq (1 - \gamma) \left\| X^t - \bar{X}^t \right\| + \gamma(1 - \delta) \left\| X^t - \bar{X}^t \right\|$$
$$= (1 - \gamma\delta) \left\| X^t - \bar{X}^t \right\|, \tag{45}$$

because $\boldsymbol{M} \left( X^t - \bar{X}^t \right) = \bar{X}^t - \bar{X}^t = 0$ and $\|\boldsymbol{A} - \boldsymbol{M}\|_2 = 1 - \delta$. The expectation of the second term can be estimated by

$$\mathbb{E}_Q \left\| -\gamma \boldsymbol{L} \left( \hat{X}^{t+1} - X^t \right) - \eta_t (\boldsymbol{I} - \boldsymbol{M}) \nabla F^t(X^t) + (\boldsymbol{I} - \boldsymbol{M}) R^{t+1} \right\|^2$$
$$\leq \mathbb{E}_Q \left( 3 \left\| \gamma \boldsymbol{L} \left( \hat{X}^{t+1} - X^t \right) \right\|^2 + 3 \left\| \eta_t (\boldsymbol{I} - \boldsymbol{M}) \nabla F^t(X^t) \right\|^2 + 3 \left\| (\boldsymbol{I} - \boldsymbol{M}) R^{t+1} \right\|^2 \right)$$
$$\overset{(39)}{\leq} 3 \left( \gamma^2 \beta^2 (1 - \omega) \left\| \hat{X}^t - X^t \right\|^2 + NG^2 \eta_t^2 + 2(1 - \omega)\beta^2 \gamma^2 \left\| X^t - \hat{X}^t \right\|^2 + 2NG^2 \eta_t^2 \right)$$
$$= 9 \left( (1 - \omega)\beta^2 \gamma^2 \left\| X^t - \hat{X}^t \right\|^2 + NG^2 \eta_t^2 \right). \tag{46}$$

By taking expectation over $Q$ w.r.t the inequality (44), together with (45) and (46), we obtain

$$\mathbb{E}_Q \left\| X^{t+1} - \bar{X}^{t+1} \right\|^2 \leq \left( 1 + \frac{\gamma\delta}{2} \right) (1 - \gamma\delta)^2 \mathbb{E}_Q \left\| X^t - \bar{X}^t \right\|^2 + 9 \left( 1 + \frac{2}{\gamma\delta} \right) NG^2 \eta_t^2$$
$$+ 9 \left( 1 + \frac{2}{\gamma\delta} \right) (1 - \omega)\beta^2 \gamma^2 \left\| X^t - \hat{X}^t \right\|^2 \tag{47}$$
$$\leq (1 - \gamma\delta)\mathbb{E}_Q \left\| X^t - \bar{X}^t \right\|^2 + 9 \left( 1 + \frac{2}{\gamma\delta} \right) NG^2 \eta_t^2$$
$$+ 9 \left( 1 + \frac{2}{\delta} \right) (1 - \omega)\beta^2 \gamma \left\| X^t - \hat{X}^t \right\|^2, \tag{48}$$

since $\left( 1 + \frac{\gamma\delta}{2} \right) (1 - \gamma\delta)^2 \leq \left( 1 - \frac{\gamma\delta}{2} \right) (1 - \gamma\delta) \leq 1 - \gamma\delta$ and $\gamma \leq 1$.

(iii) Similarly to the procedure of (ii), we have

$$\left\| X^{t+1} - \hat{X}^{t+1} \right\|^2 \overset{(28)}{=} \left\| X^t - \gamma \boldsymbol{L} \hat{X}^{t+1} - \eta_t \nabla F^t(X^t) + R^{t+1} - \hat{X}^{t+1} \right\|^2$$
$$= \left\| (\boldsymbol{I} + \gamma \boldsymbol{L}) \left( X^t - \hat{X}^{t+1} \right) - \gamma \boldsymbol{L} \left( X^t - \bar{X}^t \right) - \eta_t \nabla F^t(X^t) + R^{t+1} \right\|^2$$
$$\overset{(17)}{\leq} \left( 1 + \frac{\omega}{2} \right) \left\| (\boldsymbol{I} + \gamma \boldsymbol{L}) \left( X^t - \hat{X}^{t+1} \right) \right\|^2$$
$$+ \left( 1 + \frac{2}{\omega} \right) \left\| -\gamma \boldsymbol{L} \left( X^t - \bar{X}^t \right) - \eta_t \nabla F^t(X^t) + R^{t+1} \right\|^2. \tag{49}$$

The expectation of the first term can be estimated by

$$\mathbb{E}_Q \left\| (\boldsymbol{I} + \gamma \boldsymbol{L}) \left( X^t - \hat{X}^{t+1} \right) \right\|^2 \leq (1 + \gamma\beta)^2 \mathbb{E}_Q \left\| X^t - \hat{X}^{t+1} \right\|^2$$
$$\overset{(42)}{\leq} (1 + \gamma\beta)^2 (1 - \omega) \left\| X^t - \hat{X}^t \right\|^2, \tag{50}$$

due to $\|\boldsymbol{I} + \gamma \boldsymbol{L}\|_2 = 1 + \gamma\|L\|_2 = 1 + \gamma\beta$, since the eigenvalues of $\gamma L$ are positive. The expectation of the second term can be estimated by

$$\mathbb{E}_Q \left\| -\gamma \boldsymbol{L} \left( X^t - \bar{X}^t \right) - \eta_t \nabla F^t(X^t) + R^{t+1} \right\|^2$$
$$\leq \mathbb{E}_Q \left( 3 \left\| \gamma \boldsymbol{L} \left( X^t - \bar{X}^t \right) \right\|^2 + 3 \left\| \eta_t \nabla F^t(X^t) \right\|^2 + 3 \left\| R^{t+1} \right\|^2 \right)$$
$$\overset{(39)}{\leq} 3\gamma^2 \beta^2 \mathbb{E}_Q \left\| X^t - \bar{X}^t \right\|^2 + 3NG^2 \eta_t^2 + 6(1 - \omega)\beta^2 \gamma^2 \left\| X^t - \hat{X}^t \right\|^2 + 6NG^2 \eta_t^2. \tag{51}$$

By taking expectation over $Q$ w.r.t the inequality (49), together with (50) and (51), we obtain

$$\mathbb{E}_Q \left\| X^{t+1} - \hat{X}^{t+1} \right\|^2 \le 3 \left( 1 + \frac{2}{\omega} \right) \gamma^2 \beta^2 \mathbb{E}_Q \left\| X^t - \bar{X}^t \right\|^2 + 9 \left( 1 + \frac{2}{\omega} \right) NG^2 \eta_t^2$$

$$+ \left( \left( 1 + \frac{\omega}{2} \right) (1 - \omega)(1 + \gamma\beta)^2 + 6 \left( 1 + \frac{2}{\omega} \right) (1 - \omega)\beta^2 \gamma^2 \right) \left\| X^t - \hat{X}^t \right\|^2$$

$$\le 3 \left( 1 + \frac{2}{\omega} \right) \gamma\beta^2 \mathbb{E}_Q \left\| X^t - \bar{X}^t \right\|^2 + 9 \left( 1 + \frac{2}{\omega} \right) NG^2 \eta_t^2$$

$$+ \left( \left( 1 + \frac{\omega}{2} \right) (1 - \omega)(1 + (\beta^2 + 2\beta)\gamma) + 6 \left( 1 + \frac{2}{\omega} \right) (1 - \omega)\beta^2 \gamma \right) \left\| X^t - \hat{X}^t \right\|^2, \quad (52)$$

since $\gamma^2 \le \gamma$ for $\gamma \in (0, 1]$. $\qquad\qquad\square$

**Lemma 3.** *Suppose Assumptions 1, 2, and 4 hold. Consider Algorithm 1 with the consensus stepsize $\gamma$ chosen as (5) and arbitrary gradient descent stepsizes $\{\eta_t\}_{t=1}^T$. Define*

$$e_t := \left\| \begin{bmatrix} \mathbb{E}_Q \| X^{t+1} - \bar{X}^{t+1} \| \\ \mathbb{E}_Q \| X^{t+1} - \hat{X}^{t+1} \| \end{bmatrix} \right\|.$$

*Then for $t = 1, \cdots, T$,*

$$e_{t+1} \le \left( 1 - \frac{3}{4}\delta\gamma \right) e_t + 18 \left( 1 + \frac{1}{\gamma\delta} + \frac{1}{\omega} \right) NG^2 \eta_t^2. \qquad (53)$$

*Proof.* By Lemma 2, we have

$$\begin{bmatrix} \mathbb{E}_Q \| X^{t+1} - \bar{X}^{t+1} \| \\ \mathbb{E}_Q \| X^{t+1} - \hat{X}^{t+1} \| \end{bmatrix} \le U(\gamma) \begin{bmatrix} \mathbb{E}_Q \| X^t - \bar{X}^t \| \\ \mathbb{E}_Q \| X^t - \hat{X}^t \| \end{bmatrix} + 9NG^2 \eta_t^2 \begin{bmatrix} 1 + \frac{2}{\gamma\delta} \\ 1 + \frac{2}{\omega} \end{bmatrix}, \qquad (54)$$

where

$$U(\gamma) := \begin{bmatrix} 1 - \delta\gamma & 9 \left( 1 + \frac{2}{\delta} \right) (1 - \omega)\beta^2 \gamma \\ 3 \left( 1 + \frac{2}{\omega} \right) \beta^2 \gamma & \left( 1 + \frac{\omega}{2} \right) (1 - \omega)(1 + (\beta^2 + 2\beta)\gamma) + 6 \left( 1 + \frac{2}{\omega} \right) (1 - \omega)\beta^2 \gamma \end{bmatrix}.$$

For notation simplicity, we denote $u_1 = 9 \left( 1 + \frac{2}{\delta} \right) (1 - \omega)\beta^2$, $u_2 = 3 \left( 1 + \frac{2}{\omega} \right) \beta^2$, $u_3 = \left( 1 + \frac{\omega}{2} \right) (1 - \omega)(\beta^2 + 2\beta) + 6 \left( 1 + \frac{2}{\omega} \right) (1 - \omega)\beta^2$, and write

$$U(\gamma) = \begin{bmatrix} 1 - \delta\gamma & u_1\gamma \\ u_2\gamma & 1 - \frac{\omega}{2} - \frac{\omega^2}{2} + u_3\gamma \end{bmatrix}.$$

By the definition of $e_t$, we obtain

$$e_{t+1} \le \| U(\gamma) \|_2 \, e_t + 9NG^2 \eta_t^2 \left\| \begin{bmatrix} 1 + \frac{2}{\gamma\delta} \\ 1 + \frac{2}{\omega} \end{bmatrix} \right\|$$

$$\le \rho(U(\gamma)) e_t + 9NG^2 \eta_t^2 \left( 1 + \frac{2}{\gamma\delta} + 1 + \frac{2}{\omega} \right). \qquad (55)$$

Next, we focus on the spectrum radius of the matrix $U(\gamma)$. The characteristic polynomial of $U(\gamma)$ is

$$h(\tau) = \det \left( \tau I - U(\gamma) \right)$$

$$= \tau^2 - \left( 1 - \delta\gamma + 1 - \frac{\omega}{2} - \frac{\omega^2}{2} + u_3\gamma \right) \tau + (1 - \delta\gamma) \cdot \left( 1 - \frac{\omega}{2} - \frac{\omega^2}{2} + u_3\gamma \right) - u_1 u_2 \gamma^2.$$

Since

$$\Delta = \left( 1 - \delta\gamma + 1 - \frac{\omega}{2} - \frac{\omega^2}{2} + u_3\gamma \right)^2 - 4 \left( (1 - \delta\gamma) \cdot \left( 1 - \frac{\omega}{2} - \frac{\omega^2}{2} + u_3\gamma \right) - u_1 u_2 \gamma^2 \right)$$

$$= \left( 1 - \delta\gamma - \left( 1 - \frac{\omega}{2} - \frac{\omega^2}{2} + u_3\gamma \right) \right)^2 + 4 u_1 u_2 \gamma^2 \ge 0,$$

$$(56)$$

the equation $h(\tau) = 0$ has two roots $\tau_1$ and $\tau_2$. Since $1 - \delta\gamma + 1 - \frac{\omega}{2} - \frac{\omega^2}{2} + u_3\gamma \geq 0$,

$$\rho(U(\gamma)) = \max\{\tau_1, \tau_2\} = \frac{1}{2}\left(1 - \delta\gamma + 1 - \frac{\omega}{2} - \frac{\omega^2}{2} + u_3\gamma + \sqrt{\Delta}\right). \tag{57}$$

When

$$\gamma \leq \frac{2\delta(\omega^2 + \omega)}{16u_1 u_2 + 4u_3\delta + 3\delta^2}, \tag{58}$$

it can be verified that

$$\Delta \leq \left(1 - \frac{\gamma\delta}{2} - \left(1 - \frac{\omega}{2} - \frac{\omega^2}{2} + u_3\gamma\right)\right)^2, \tag{59}$$

and then,

$$\rho(U(\gamma)) \leq \frac{1}{2}\left(\left(1 - \gamma\delta + \left(1 - \frac{\omega}{2} - \frac{\omega^2}{2} + u_3\gamma\right)\right) + \left(1 - \frac{\gamma\delta}{2} - \left(1 - \frac{\omega}{2} - \frac{\omega^2}{2} + u_3\gamma\right)\right)\right)$$
$$= 1 - \frac{3}{4}\gamma\delta. \tag{60}$$

We take

$$\gamma = \gamma(\omega) := \frac{3\delta}{4}\frac{2\delta(\omega^2 + \omega)}{16u_1 u_2 + 4u_3\delta + 3\delta^2}$$
$$= \frac{3\delta^3\omega^2(\omega + 1)}{48(\delta^2 + 18\delta\beta^2 + 36\beta^2)\beta^2(\omega + 2)(1 - \omega) + 4\delta^2(\beta^2 + \beta)(\omega + 2)(1 - \omega)\omega + 6\delta^3\omega}, \tag{61}$$

which satisfies (58) since $\frac{3\delta}{4} \leq 1$. Notice that $\gamma(\omega)$ increases monotonically with $\omega$, and $\gamma(0) = 0, \gamma(1) = 1$. Thus, $\gamma(\omega) \in (0, 1]$ for $\omega \in (0, 1]$, which meets the algorithm design requirement. Then, the lemma is proved. $\qquad\square$

**Lemma 4.** *Let* $\{e_t\}_{t \geq 1}$ *denotes a sequence of real values satisfying* $e_1 = 0$ *and*

$$e_{t+1} \leq (1 - p)e_t + q\eta_t^2, \tag{62}$$

*for parameters* $p \in (0, 1)$, $q > 0$, *and the stepsize sequence* $\{\eta_t\}_{t \geq 1}$ *satisfying either of the following conditions*

*(i)* $\eta_t = \frac{b}{\sqrt{t+c}}$ *for constants* $c \geq \frac{2}{p}, b \geq 0$,

*(ii)* $\eta_t = \frac{b}{t+c}$ *for constants* $c \geq \frac{4}{p}, b \geq 0$.

*Then for any* $t \geq 1$,

$$e_t \leq \frac{2q}{p}\eta_t^2. \tag{63}$$

*Proof.* We proceed the proof by induction. For $t = 1$, the statement holds since $e_1 = 0$. Suppose that the statement holds for $t$. Then for $t + 1$,

$$e_{t+1} \leq (1 - p)e_t + q\eta_t^2 \leq (1 - p)\frac{2q}{p}\eta_t^2 + q\eta_t^2. \tag{64}$$

It remains to prove

$$(1 - p)\frac{2q}{p}\eta_t^2 + q\eta_t^2 \leq \frac{2q}{p}\eta_{t+1}^2. \quad \left(\Longleftrightarrow \quad 1 - \frac{p}{2} \leq \frac{\eta_{t+1}^2}{\eta_t^2}\right) \tag{65}$$

As for the condition (i),

$$\frac{\eta_{t+1}^2}{\eta_t^2} = \frac{t + c}{t + c + 1} = 1 - \frac{1}{t + c + 1} > 1 - \frac{p}{2}, \quad \forall t \geq 1. \tag{66}$$

As for the condition (ii),

$$\frac{\eta_{t+1}^2}{\eta_t^2} = \left(\frac{t + c}{t + c + 1}\right)^2 > 1 - \frac{2}{t + c + 1} > 1 - \frac{p}{2}, \quad \forall t \geq 1. \tag{67}$$

Thus, the conclusion follows. $\qquad\square$

**Proof of Theorem 1**

By Lemma 3 and Lemma 4, we have

$$e_t \leq \frac{48}{\gamma\delta}\left(1 + \frac{1}{\gamma\delta} + \frac{1}{\omega}\right)NG^2\eta_t^2. \tag{68}$$

According to the Jensen's Inequality,

$$\mathbb{E}_Q\|X^t - \bar{X}^t\| \overset{(21)}{\leq} \sqrt{\mathbb{E}_Q\|X^t - \bar{X}^t\|^2} \leq \sqrt{e_t} \leq \sqrt{\frac{48}{\gamma\delta}\left(1 + \frac{1}{\gamma\delta} + \frac{1}{\omega}\right)NG^2\eta_t^2}$$

$$\leq 4\sqrt{3}\left(1 + \frac{1}{\gamma\delta} + \frac{1}{\omega}\right)\sqrt{N}G\eta_t. \tag{69}$$

Similarly to the procedure of (44), we estimate

$$\left\|\bar{X}^t - \widetilde{X}^{t+1}\right\|^2 = \left\|X^t - \gamma\boldsymbol{L}\hat{X}^{t+1} - \eta_t\nabla F^t(X^t) - \bar{X}^t\right\|^2$$

$$= \left\|(\boldsymbol{I} - \gamma\boldsymbol{L})\left(X^t - \bar{X}^t\right) - \gamma\boldsymbol{L}\left(\hat{X}^{t+1} - X^t\right) - \eta_t\nabla F^t(X^t)\right\|^2$$

$$\overset{(17)}{\leq} \left(1 + \frac{\gamma\delta}{2}\right)\left\|(\boldsymbol{I} - \gamma\boldsymbol{L})\left(X^t - \bar{X}^t\right)\right\|^2 \tag{70}$$

$$+ \left(1 + \frac{2}{\gamma\delta}\right)\left((1 + 2)\left\|\gamma\boldsymbol{L}\left(\hat{X}^{t+1} - X^t\right)\right\|^2 + \left(1 + \frac{1}{2}\right)\left\|\eta_t\nabla F^t(X^t)\right\|^2\right).$$

Together with the estimate of $\mathbb{E}_Q\|R^{t+1}\|^2$ and the choice of $\gamma$ in (5), we have

$$\mathbb{E}_Q\left(\left\|\bar{X}^t - \widetilde{X}^{t+1}\right\|^2 + 3\|R^{t+1}\|^2\right)$$

$$\leq \left(1 + \frac{\gamma\delta}{2}\right)(1 - \gamma\delta)^2\mathbb{E}_Q\left\|X^t - \bar{X}^t\right\|^2 + 3\left(1 + \frac{2}{\gamma\delta}\right)(1 - \omega)\beta^2\gamma^2\left\|X^t - \hat{X}^t\right\|^2$$

$$+ \frac{3}{2}\left(1 + \frac{2}{\gamma\delta}\right)NG^2\eta_t^2 + 3\left(2(1 - \omega)\beta^2\gamma^2\left\|X^t - \hat{X}^t\right\|^2 + 2NG^2\eta_t^2\right)$$

$$\leq (1 - \gamma\delta)\mathbb{E}_Q\left\|X^t - \bar{X}^t\right\|^2 + 9\left(1 + \frac{2}{\delta}\right)(1 - \omega)\beta^2\gamma\left\|X^t - \hat{X}^t\right\|^2 + \left(\frac{15}{2} + \frac{3}{\gamma\delta}\right)NG^2\eta_t^2$$

$$= \begin{bmatrix} 1 & 0 \end{bmatrix}U(\gamma)\begin{bmatrix}\mathbb{E}_Q\|X^t - \bar{X}^t\| \\ \mathbb{E}_Q\|X^t - \hat{X}^t\|\end{bmatrix} + \left(\frac{15}{2} + \frac{3}{\gamma\delta}\right)NG^2\eta_t^2$$

$$\leq \rho(U(\gamma))e_t + \left(\frac{15}{2} + \frac{3}{\gamma\delta}\right)NG^2\eta_t^2$$

$$\overset{(68)}{\leq} \left(1 - \frac{3}{4}\gamma\delta\right)\frac{48}{\gamma\delta}\left(1 + \frac{1}{\gamma\delta} + \frac{1}{\omega}\right)NG^2\eta_t^2 + \left(\frac{15}{2} + \frac{3}{\gamma\delta}\right)NG^2\eta_t^2$$

$$= \left(\left(\frac{48}{\gamma\delta} - 36\right)\left(1 + \frac{1}{\gamma\delta} + \frac{1}{\omega}\right) + \frac{15}{2} + \frac{3}{\gamma\delta}\right)NG^2\eta_t^2, \tag{71}$$

where $U(\gamma)$ is defined in Lemma 3.

For the strongly convex case (ii), we substitute (69) and (71) into (30), and derive

$$\mathbb{E}_Q\,\mathrm{R}(j, T) \leq \mu c D^2 + NG^2\sum_{t=1}^{T}\eta_t + (2\sqrt{N} + N)G\sum_{t=1}^{T}4\sqrt{3}\left(1 + \frac{1}{\gamma\delta} + \frac{1}{\omega}\right)\sqrt{N}G\eta_t$$

$$+ \sum_{t=1}^{T}\frac{1}{2\eta_t}\left(\left(\frac{48}{\gamma\delta} - 36\right)\left(1 + \frac{1}{\gamma\delta} + \frac{1}{\omega}\right) + \frac{15}{2} + \frac{3}{\gamma\delta}\right)NG^2\eta_t^2$$

$$\leq \mu c D^2 + 4\sqrt{3}\left(\sqrt{N} + \frac{2\sqrt{3}}{\gamma\delta} + 1\right)\left(1 + \frac{1}{\gamma\delta} + \frac{1}{\omega}\right)NG^2\sum_{t=1}^{T}\eta_t, \tag{72}$$

where the last inequality holds since $\left(4\sqrt{3} + \frac{3}{2} - 18\right)/\gamma\delta < 0$ and $1 + (4\sqrt{3} - 18)(1 + \frac{1}{\omega}) + \frac{15}{4} < 0$ for $\omega \in (0, 1]$. Consider the gradient stepsize $\eta_t = \frac{1}{\mu(t+c)}$ for $c \geq \frac{16}{3\gamma\delta} > 1$, and then,

$$\sum_{t=1}^{T} \eta_t = \sum_{t=1}^{T} \frac{1}{\mu(t+c)} \leq \frac{1}{\mu} \int_0^T \frac{1}{s+c} ds = \frac{1}{\mu} \ln(s+c)|_0^T \leq \frac{1}{\mu} \ln(T+c). \tag{73}$$

Combining (72) with (73), we obtain (7).

For the convex case (i), the gradient stepsize $\eta_t = \frac{D}{G\sqrt{t+c}}$, and then,

$$\sum_{t=1}^{T} \eta_t = \sum_{t=1}^{T} \frac{D}{G\sqrt{t+c}} \leq \frac{D}{G} \int_0^T \frac{1}{\sqrt{s+c}} ds = \frac{D}{G} 2\sqrt{s+c}|_0^T \leq \frac{2D}{G}\sqrt{T+c}. \tag{74}$$

By substituting (69), (71), and (74) into (29), we derive

$$\mathbb{E}_Q \, \mathrm{R}(j, T) \leq \frac{ND^2}{2\eta_T} + 4\sqrt{3}\left(\sqrt{N} + \frac{2\sqrt{3}}{\gamma\delta} + 1\right)\left(1 + \frac{1}{\gamma\delta} + \frac{1}{\omega}\right) NG^2 \sum_{t=1}^{T} \eta_t$$

$$\leq \frac{ND^2}{2} \frac{G\sqrt{T+c}}{D} + 4\sqrt{3}\left(\sqrt{N} + \frac{2\sqrt{3}}{\gamma\delta} + 1\right)\left(1 + \frac{1}{\gamma\delta} + \frac{1}{\omega}\right) NG^2 \frac{2D}{G}\sqrt{T+c}.$$

Then the theorem is proved. $\qquad\square$

## C  Proofs of Section 3

Algorithm 2 actually performs the gradient descent scheme on the function $\hat{f}_i^t(x) = \mathbb{E}_{u\in\mathcal{B}}\left[f_i^t(x + \epsilon u)\right]$ restricted to the convex set $(1 - \zeta)\mathcal{K}$. By Assumptions 6 and 7, as well as the construction of $g_i^t$,

$$\left\|\nabla \hat{f}_i^t(x)\right\| = \left\|\mathbb{E}\left[g_i^t\right]\right\| \leq \mathbb{E}\left[\|g_i^t\|\right] \leq \mathbb{E}\left[\frac{d}{\epsilon}\|f_i^t\|\|u_i^t\|\right] \leq \frac{dB}{\epsilon} := G, \quad \forall i \in \mathcal{V}, t = 1, \cdots, T,$$

$$\|x - y\| \leq 2R := D, \quad \forall x, y \in (1 - \zeta)\mathcal{K}.$$

The remaining gaps include 1) the difference between the case of the loss function $f_i^t$ and that of $\hat{f}_i^t$; 2) the difference between the case of the feasible set $(1 - \zeta)\mathcal{K}$ and that of $\mathcal{K}$. As for 1), by Assumption 7, we have

$$\left\|\hat{f}_i^t(x) - f_i^t(x)\right\| = \left\|\mathbb{E}_u\left[f_i^t(x + \epsilon u)\right] - f_i^t(x)\right\| \leq \mathbb{E}_u\left\|f_i^t(x + \epsilon u) - f_i^t(x)\right\| \leq l\epsilon,$$

and thus,

$$\hat{f}_i^t(x) - l\epsilon \leq f_i^t(x) \leq \hat{f}_i^t(x) + l\epsilon. \tag{75}$$

As for 2), we have the following lemma from [4].

**Lemma 5.** *The optimum in* $(1 - \zeta)\mathcal{K}$ *is near the optimum in* $\mathcal{K}$.

$$\min_{x\in(1-\zeta)\mathcal{K}} \sum_{t=1}^{T}\sum_{i=1}^{N} f_i^t(x) \leq 2\zeta BNT + \min_{x\in\mathcal{K}} \sum_{t=1}^{T}\sum_{i=1}^{N} f_i^t(x). \tag{76}$$

By Lemma 5, we can obtain the regret bounds in the one-point bandit setting upon the obtained results in the full information setting.

$$\mathbb{E}\left[\mathrm{R}(j, T)\right] = \mathbb{E}\sum_{t=1}^{T}\sum_{i=1}^{N} f_i^t(x_j^t) - \min_{x\in\mathcal{K}} \sum_{t=1}^{T}\sum_{i=1}^{N} f_i^t(x)$$

$$\overset{(76)}{\leq} \mathbb{E}\sum_{t=1}^{T}\sum_{i=1}^{N} f_i^t(x_j^t) - \min_{x\in(1-\zeta)\mathcal{K}} \sum_{t=1}^{T}\sum_{i=1}^{N} f_i^t(x) + 2\zeta BNT$$

$$\overset{(75)}{\leq} \mathbb{E}\sum_{t=1}^{T}\sum_{i=1}^{N} \left(\hat{f}_i^t(x_j^t) + l\epsilon\right) - \min_{x\in(1-\zeta)\mathcal{K}} \sum_{t=1}^{T}\sum_{i=1}^{N} \left(\hat{f}_i^t(x_j^t) - l\epsilon\right) + 2\zeta BNT$$

$$= \mathbb{E}\sum_{t=1}^{T}\sum_{i=1}^{N} \hat{f}_i^t(x_j^t) - \min_{x\in(1-\zeta)\mathcal{K}} \sum_{t=1}^{T}\sum_{i=1}^{N} \hat{f}_i^t(x_j^t) + 2l\epsilon NT + 2\zeta BNT. \tag{77}$$

**Proof of Theorem 2**

(i) (Convex case) From Theorem 1 part (i), we have

$$\mathbb{E}\sum_{t=1}^{T}\sum_{i=1}^{N}\hat{f}_i^t(x_j^t) - \min_{x\in(1-\zeta)\mathcal{K}}\sum_{t=1}^{T}\sum_{i=1}^{N}\hat{f}_i^t(x) \le \left(\frac{1}{2}+2H\right)N\frac{dB}{\epsilon}2R\sqrt{T+c}, \tag{78}$$

where $H$ is defined in (8). Then by (77) with $\zeta = \frac{\epsilon}{r}$,

$$\mathbb{E}\left[\mathrm{R}(j,T)\right] \le (1+4H)\,N\frac{dBR}{\epsilon}\sqrt{T+c} + 2l\epsilon NT + 2\frac{\epsilon}{r}BNT. \tag{79}$$

We choose $\epsilon = \left(\frac{(1+4H)dBR}{2\left(l+\frac{B}{r}\right)}\right)^{\frac{1}{2}}\frac{(T+c)^{\frac{1}{4}}}{T^{\frac{1}{2}}}$ to minimize the right hand of the above inequality and then obtain the conclusion.

(ii) (Strongly convex case) From Theorem 1 part (ii), we have

$$\mathbb{E}\sum_{t=1}^{T}\sum_{i=1}^{N}\hat{f}_i^t(x_j^t) - \min_{x\in(1-\zeta)\mathcal{K}}\sum_{t=1}^{T}\sum_{i=1}^{N}\hat{f}_i^t(x) \le 4\mu cR^2 + H\frac{Nd^2B^2}{\mu\epsilon^2}\ln(T+c), \tag{80}$$

where $H$ is defined in (8). Then by (77) with $\zeta = \frac{\epsilon}{r}$,

$$\mathbb{E}\left[\mathrm{R}(j,T)\right] \le 4\mu cR^2 + H\frac{Nd^2B^2}{\mu\epsilon^2}\ln(T+c) + 2l\epsilon NT + 2\frac{\epsilon}{r}BNT. \tag{81}$$

We choose $\epsilon = \left(\frac{\frac{Hd^2B^2}{\mu}\ln(T+c)}{\left(l+\frac{B}{r}\right)T}\right)^{\frac{1}{3}}$ to minimize the right hand of the above inequality and then obtain the conclusion. $\qquad\square$

# D Proofs of Section 4

The proof in the two-point bandit case takes a similar procedure as that in the one-point case. By Assumptions 6 and 7, as well as the construction of $g_i^t$,

$$\left\|\nabla\hat{f}_i^t(x)\right\| = \left\|\mathbb{E}\left[g_i^t\right]\right\| \le \mathbb{E}\left[\|g_i^t\|\right] \le \mathbb{E}\left[\frac{d}{2\epsilon}\left\|f_i^t(x_i^t+\epsilon u_i^t) - f_i^t(x_i^t-\epsilon u_i^t)\right\|\left\|u_i^t\right\|\right]$$

$$\le \frac{d}{2\epsilon}l2\epsilon\left\|u_i^t\right\|^2 = dl := G, \quad \forall i\in\mathcal{V}, t=1,\cdots,T,$$

$$\|x-y\| \le 2R := D, \quad \forall x,y\in(1-\zeta)\mathcal{K}.$$

Similar to the Lemma 2 in [5], we have

**Lemma 6.** *For any point* $x\in\mathcal{K}$,

$$\sum_{t=1}^{T}\sum_{i=1}^{N}\frac{f_i^t(y_{i,1}^t) + f_i^t(y_{i,2}^t)}{2} - \sum_{t=1}^{T}\sum_{i=1}^{N}f_i^t(x)$$

$$\le \sum_{t=1}^{T}\sum_{i=1}^{N}\hat{f}_i^t(x_j^t) - \sum_{t=1}^{T}\sum_{i=1}^{N}\hat{f}_i^t((1-\zeta)x) + 3NTG\epsilon + NTGD\zeta. \tag{82}$$

By Lemma 6, for $x^* = \arg\min\limits_{x\in\mathcal{K}}\sum_{t=1}^{T}\sum_{i=1}^{N}f_i^t(x)$,

$$\mathbb{E}\left[\mathrm{R}_2(j,T)\right] = \mathbb{E}\sum_{t=1}^{T}\sum_{i=1}^{N}\frac{f_i^t(y_{i,1}^t) + f_i^t(y_{i,2}^t)}{2} - \sum_{t=1}^{T}\sum_{i=1}^{N}f_i^t(x^*)$$

$$\le \mathbb{E}\sum_{t=1}^{T}\sum_{i=1}^{N}\hat{f}_i^t(x_j^t) - \sum_{t=1}^{T}\sum_{i=1}^{N}\hat{f}_i^t((1-\zeta)x^*) + 3NTG\epsilon + NTGD\zeta$$

$$\le \mathbb{E}\sum_{t=1}^{T}\sum_{i=1}^{N}\hat{f}_i^t(x_j^t) - \min_{x\in(1-\zeta)\mathcal{K}}\sum_{t=1}^{T}\sum_{i=1}^{N}\hat{f}_i^t(x) + 3NTdl\epsilon + 2NTdlR\zeta \tag{83}$$

**Proof of Theorem 3**

(i) (Convex case) From Theorem 1 part (i), we have

$$\mathbb{E}\sum_{t=1}^{T}\sum_{i=1}^{N}\hat{f}_i^t(x_j^t) - \min_{x\in(1-\varsigma)\mathcal{K}}\sum_{t=1}^{T}\sum_{i=1}^{N}\hat{f}_i^t(x) \le \left(\frac{1}{2}+2H\right)Ndl2R\sqrt{T+c}, \tag{84}$$

where $H$ is defined in (8). Then by (83) with $\zeta = \frac{\epsilon}{r}$,

$$\mathbb{E}\left[R_2(j,T)\right] \le (1+4H)\,NdlR\sqrt{T+c} + \left(3+\frac{2R}{r}\right)NdlT\epsilon. \tag{85}$$

We choose $\epsilon = \frac{1}{\sqrt{T}}$ and then obtain the conclusion.

(ii) (Strongly convex case) From Theorem 1 part (ii), we have

$$\mathbb{E}\sum_{t=1}^{T}\sum_{i=1}^{N}\hat{f}_i^t(x_j^t) - \min_{x\in(1-\varsigma)\mathcal{K}}\sum_{t=1}^{T}\sum_{i=1}^{N}\hat{f}_i^t(x) \le 4\mu c R^2 + H\frac{Nd^2l^2}{\mu}\ln(T+c), \tag{86}$$

where $H$ is defined in (8). Then by (83) with $\zeta = \frac{\epsilon}{r}$,

$$\mathbb{E}\left[R_2(j,T)\right] \le 4\mu c R^2 + H\frac{Nd^2l^2}{\mu}\ln(T+c) + \left(3+\frac{2R}{r}\right)NdlT\epsilon \tag{87}$$

We choose $\epsilon = \frac{\ln(T)}{T}$ and then obtain the conclusion. $\qquad\square$

# E    Parameters selection details

The theoretical value of the consensus stepsize $\gamma$ depends on the compression ratio $\omega$ and the graph parameters $\delta, \beta$, which is pretty conservative. We tune $\gamma$ for each experiment. The gradient descent stepsizes of DC-DOGD, DC-DOBD and DC-DO2BD for convex losses can be written in a unified form as $\eta_t = \frac{b}{\sqrt{t+c}}$, where $c = \frac{8}{3\gamma\delta}$ as in theorems and $b$ is tuned from $\{0.001, 0.005, 0.01, 0.05, 0.1, 0.5, 1\}$. Also, for strongly convex losses, the gradient descent stepsize can be written as $\eta_t = \frac{b}{t+c}$, where $c = \frac{16}{3\gamma\delta}$ as in theorems and $b$ is tuned. In DC-DOBD and DC-DO2BD, the shrinkage parameter $\zeta = \frac{\epsilon}{r}$ as in theorems and the exploration parameter $\epsilon$ is tuned from $\{0.001, 0.005, 0.01, 0.05, 0.1, 0.5, 1\}$. For each experiment, $b$ and $\epsilon$ are tuned by grid search.

**Parameters in Fig. 1**    In this experiment, we set $\gamma = 0.26$ for $\mathrm{QSGD}_2$ with $\omega = 0.3$ over $\mathcal{G}(9,18)$. The parameters $b$ and $\epsilon$ for the proposed algorithms are given in Table 2. The parameters for ECD-AMDGrad are chosen as suggested in [6].

Table 2: Parameters $b$ and $\epsilon$ for the proposed algorithms

| Parameters | Convex losses | | Strongly convex | |
|---|---|---|---|---|
| | $b$ | $\epsilon$ | $b$ | $\epsilon$ |
| DC-DOGD | 0.1 | \ | 1 | \ |
| DC-DOBD | 0.01 | 0.5 | 0.05 | 0.5 |
| DC-DO2BD | 0.1 | 0.05 | 0.5 | 0.01 |

**Parameters in Fig. 2**    When studying the impacts of compression ratio and compressor type, we take DC-DOGD with strongly convex losses over the graph $\mathcal{G}(N, 2N)$ as an example, and set $b = 1$. The corresponding $\gamma$ for different compression ratios $w$ (with the same compressor type $\mathrm{Top}_k$) are given in Table 3, and the corresponding $\gamma$ for different compressor types (with the same compression ratio $\omega = 0.3$) are given in Table 4. DAOL takes the same gradient descent stepsizes as DC-DOGD.

**Parameters in Fig. 3**    When studying the impact of network topology, we take DC-DOGD with strongly convex losses as an example, and set $b = 1$. For the compressor $\mathrm{Top}_1$ with $\omega = 0.05$, we set $\gamma = 0.09$. When studying the impact of node number, we take the compressor $\mathrm{Top}_2$ with $\omega = 0.1$ and set $\gamma = 0.1$. The parameters $b$ and $\epsilon$ are chosen the same as Table 2.

Table 3: Corresponding $\gamma$ for different $\omega$

| $\omega$ | 0.05 | 0.1 | 0.5 |
|---|---|---|---|
| $\gamma$ | 0.09 | 0.1 | 0.32 |

Table 4: Corresponding $\gamma$ for different compressors

| Compressor | $\mathrm{RGossip}_p$ | $\mathrm{Rand}_k$ | $\mathrm{Top}_k$ | $\mathrm{GSGD}_s$ |
|---|---|---|---|---|
| $\gamma$ | 0.09 | 0.09 | 0.28 | 0.26 |

## F  Additional experiments

We give some additional experiments in the convex cases here. Still, we use the dataset *diabetes-binary-BRFSS2015* [5]. The communication graph is generated by the tool *NetworkX* [6] and the best solution is obtained by the tool *Logistic Regression* [7]. Our code is available at `https://github.com/happy-math/CC-DOCO`.

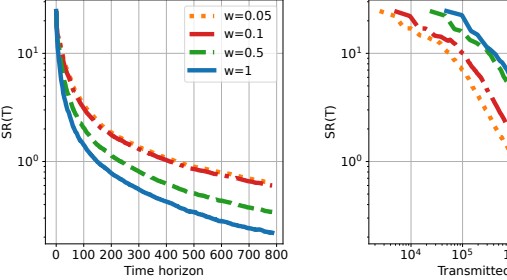

Figure 4: The impact of compression ratio $\omega$. Setting: DC-DOGD with the compressor $\mathrm{Top}_k$ over $\mathcal{G}(9, 18)$ in the convex case. $b = 1$. The corresponding $\gamma$ for different $w$ are chosen as in Table 3.

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

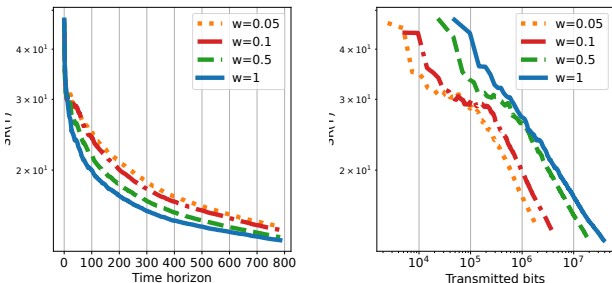

Figure 5: The impact of compression ratio $\omega$. Setting: DC-DOBD with the compressor $\text{Top}_k$ over $\mathcal{G}(9, 18)$ in the convex case. $b = 0.01, \epsilon = 1$. The corresponding $\gamma$ for different $w$ are chosen as in Table 3.

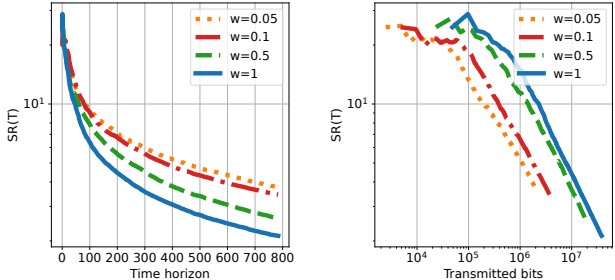

Figure 6: The impact of compression ratio $\omega$. Setting: DC-DO2BD with the compressor $\text{Top}_k$ over $\mathcal{G}(9, 18)$ in the convex case. $b = 0.1, \epsilon = 0.05$. The corresponding $\gamma$ for different $w$ are chosen as in Table 3.

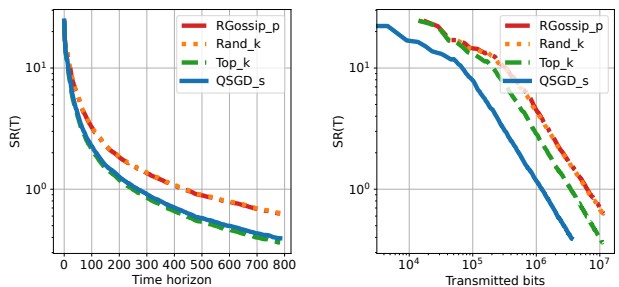

Figure 7: The impact of compressor type. Setting: DC-DOGD with the compression ration $\omega = 0.3$ over $\mathcal{G}(9, 18)$ in the convex case. $b = 0.1$. The corresponding $\gamma$ for different compressor types are chosen as in Table 4.

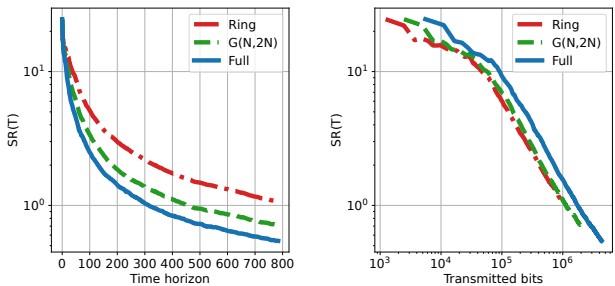

Figure 8: The impact of topology. Setting: DC-DOGD with $\text{Top}_1, \omega = 0.05$ in the convex case. $b = 0.1, \gamma = 0.09$.