# OpenReview forum: "Distributed Online Convex Optimization with Compressed Communication"
_NeurIPS.cc/2022/Conference — NeurIPS 2022 Accept_

### Official Review · Reviewer_wakL · 2022-07-08

**Rating:** 3
**Confidence:** 5
**Soundness:** 3 good
**Presentation:** 2 fair
**Contribution:** 2 fair

**Summary:**

This work is an augmentation of distributed online gradient descent that uses an update-difference based compression scheme to reduce the communications load. Sublinear regret is proved for the case that objective functions are convex and strongly convex. Numerical evaluations establish the practicalities of the proposed approach.

**Questions:**

How do the developed regret bounds here compare against the tightest possible bounds in the literature? How is the compression ratio related to error in the gradient estimation error of the Lagrangian of the consensus-constrained optimization problem? How is it related to the error bound condition that is well-known in the mathematical programming literature (see many of Paul Tseng's works)?

Luo, Z. Q., & Tseng, P. (1993). Error bounds and convergence analysis of feasible descent methods: a general approach. Annals of Operations Research, 46(1), 157-178.

Nedic, A., Ozdaglar, A., & Parrilo, P. A. (2010). Constrained consensus and optimization in multi-agent networks. IEEE Transactions on Automatic Control, 55(4), 922-938.

**Limitations:**


The authors propose to employ difference compression together with DGD in order to come up with a communication-efficient algorithm. What I do not see is any technical novelty beyond applying this compression technique to a previously existing setting. Put more succinctly, there is no specific algorithmic innovation beyond applying a previously compression technique to a previously existing setting, and analyzing the error incurred by compression. For this reason, I do not believe this paper meets the bar for technical creativity/novelty required for NeurIPS.


**Strengths And Weaknesses:**


The numerical experiments only consider weak baselines, in the sense that they do not compare against ADMM or primal-dual method. Perhaps this is not the point, as the authors would like to underscore that the difference based compression is the best among a collection of alternatives. However, ECD-AMSGrad is a weak comparator, and many communication-efficient versions of ADMM or primal-dual method are available in the literature. See, for instance:

Ling, Qing, Yaohua Liu, Wei Shi, and Zhi Tian. "Weighted ADMM for fast decentralized network optimization." IEEE Transactions on Signal Processing 64, no. 22 (2016): 5930-5942.

Bedi, A. S., Koppel, A., & Rajawat, K. (2019). Asynchronous saddle point algorithm for stochastic optimization in heterogeneous networks. IEEE Transactions on Signal Processing, 67(7), 1742-1757.

Lines 29-38: The authors only talk about local and global regret. However, in this setting, one should also talk about the tradeoff between global regret and constraint violation in the sense of consensus error, so as to make the link with primal-dual method and ADMM, which have rigorous and extensive theoretical support that they are superior to distributed gradient descent-based techniques. That this line of literature is overlooked completely so that a DOGD based development can be pursued is evidence to me that the paper exhibits some substantial gaps between the proposed algorithm and the state of the art.

Koppel, A., Jakubiec, F. Y., & Ribeiro, A. (2015). A saddle point algorithm for networked online convex optimization. IEEE Transactions on Signal Processing, 63(19), 5149-5164.

Boyd, S., Parikh, N., Chu, E., Peleato, B., & Eckstein, J. (2011). Distributed optimization and statistical learning via the alternating direction method of multipliers. Foundations and Trends® in Machine learning, 3(1), 1-122.

Moreover, there are specific corrections of DGD that close the gap between it and approaches based on duality, which are also overlooked:

Shi, Wei, et al. "Extra: An exact first-order algorithm for decentralized consensus optimization." SIAM Journal on Optimization 25.2 (2015): 944-966.


Line 41: There are much more related works than [7]-[9] in the study of decentralized online convex optimization. Moreover, references [8] and [9] appear to be about constrained settings which are not particularly pertinent here.

---

> ### Author Response · Authors · 2022-08-02
> **Response to the reviewer wakL (Part 2/2)**
>
> ```
> Q5: How do the developed regret bounds here compare against the tightest possible bounds in the literature?
> ```
> For convex and strongly convex losses with full gradient feedback and two-point information, $\mathcal O(\sqrt T)$ and $\mathcal O(\operatorname {ln}(T))$ regrets are established, respectively, which are optimal bounds and match those obtained by the state-of-art distributed online algorithms.
>
> For convex and strongly convex losses with one-point information, $\mathcal O(T^{\frac{3}{4}})$ and $\mathcal O (T^{\frac{2}{3}}\operatorname{ln}^{\frac{1}{3}}(T))$ regrets are established, respectively, which match those obtained by the corresponding centralized online algorithms, while they are not the tightest. Better regret bounds can be achieved if additional assumptions are made. Refer to [3] and references therein.
>
> [3] Li, X., Xie, L., \& Li, N. (2022). A Survey of Decentralized Online Learning. arXiv preprint arXiv:2205.00473.
>
> ```
> Q6: How is the compression ratio related to error in the gradient estimation error of the Lagrangian of the consensus-constrained optimization problem?
> ```
> Online optimization is not concerned with the gradient estimation error. The key difference between online and offline optimization is that at each round the loss function $f_i^t$ can be arbitrarily chosen by the adversary and is revealed to node $i$ after node $i$ makes decision $x_i^t$. Without a model imposed on the function choices, the gradient cannot be estimated or controlled.
>
> ```
> Q7: How is it related to the error bound condition that is well-known in the mathematical programming literature?
> ```
> To my knowledge, the regret bound under the error bound condition is still an open question. The conditions we consider are convex and strongly-convex, which are mild and common in distributed online optimization [18]. The purpose of our paper is to address the communication bottleneck, not to make the conditions weaker. We may study it in the future.
>
> ```
> Q8: There is no specific algorithmic innovation beyond applying a previously compression technique to a previously existing setting, and analyzing the error incurred by compression.
> ```
> Although the idea of such a combination looks simple, the underlying algorithm design and theoretical principle are challenging. In the offline distributed consensus optimization, gossip consensus and gradient descent can be done whoever first or be done simultaneously. All these three manners can lead to provable convergence. However, in the distributed online optimization with compressors, the order matters. There are two key points to design provably no-regret distributed online algorithms that work with compressors.
>
> 1. Let the noise introduced by the compression vanish. Note that directly compressing the local state variable and spreading it will lead to the algorithm oscillation or even divergence. By allowing nodes to add replicas of neighboring states and compress the state-difference, we can address this problem. The replica $\hat x_i^{t+1}$ actually track the true state $x_i^t$. Then comes the next problem:
> 2. Bound the regret. If we first do the gradient descent, followed with the difference-compressed communication and finally the gossip consensus, as [14] and [18] did, then the regrets will be really difficult to estimate.
> Technically, in the traditional proof scheme, $\sum_{j=1}^Na_{ij}x_i^t\in\mathcal K$ since $x_i^t\in\mathcal K$ and $\sum_{j=1}^Na_{ij}=1$, while we do not have $\sum_{j=1}^Na_{ij}\hat x_i^{t+1}$ in $\mathcal K$ although $\hat x_i^{t+1}$ tracks $x_i^t$. This small difference results in big difficulty in the projection error estimation, which is essential for regret analysis. We solve this by carefully adjusting the algorithm design. We first do the difference-compressed communication (steps 2-4), and then do the gossip consensus and gradient descent simultaneously (step 5). By the projection property, we can carry out the estimation in Line 530 in Appendix, which successfully connects the projection error with the tracking error and stepsize. Besides, there are many tricks such as designing proper consensus stepsize to control the overall errors. One of the strengths of this paper is the proof novelty.
>
> Therefore, we believe that our results on distributed communication-efficient online optimization still contain several non-trivial developments.

---

> ### Author Response · Authors · 2022-08-02
> **Response to the reviewer wakL (Part 1/2)**
>
> Thank you for your time. We make responses to all your concerns and questions, and sincerely hope you can rethink our paper for possible publication.
>
> First of all, we would like to highlight that the problem we consider is the distributed/decentralized online optimization, where the objective functions are streaming and time-varying (unlike the traditional offline optimization) and the multi-agent network has no central agent (unlike federated learning). Our goal is to design provably no-regret distributed online algorithms that work with compressors.
> ```
> Q1: Not compare against ADMM, primal-dual method, but ECD-AMSGrad
> ```
> Although there are many communication-efficient versions of ADMM or primal-dual method (as you have mentioned) as well as many online versions of ADMM or primal-dual method, there has been no work on provable online distributed communication-efficient ADMM or primal-dual method, to the best of our knowledge. ECD-AMSGrad is one of the few works that design a  communication-efficient distributed online algorithm, by extending the AMSGrad to the distributed online setting with compressors, and thus, ECD-AMSGrad is our only comparator.
>
> ```
> Q2: One should also talk about the tradeoff between global regret and constraint violation in the sense of consensus error, so as to make the link with primal-dual method and ADMM.
> ```
> Actually, we have given the relationship between the regret and the consensus error in Appendix because of space limitation. We first bound the local regret $\operatorname R(j,T)$ on each node by the consensus error $\\|X^t-\bar X^t\\|^2$, the projection error $\\|X^t-\widetilde X^t\\|^2$, the increment $\\|\widetilde X^{t+1}-\bar X^t\\|^2$, and the stepsize $\eta_t$, in Lemma 1. Then, in Lemma 2 we analyze the coupled relationship between the errors. By Lemmas 3 and 4, we give the estimation of the consensus error as shown in Eq. (65). We will also add these description in the main manunscript.
>
> As for the link with ADMM and primal-dual method, our online GD-based algorithm achieves $\mathcal O(\sqrt T)$ regret bound for convex losses, which matches the regrets obtained by the online ADMM [1] and online primal-dual method [2] with respect to the time horizon $T$. In other words, although ADMM and primal-dual may be superior to distributed GD-based algorithms in offline optimization, they have similar regret performance in online optimization.
>
> [1] Akbari, M., Gharesifard, B., \& Linder, T. (2018). Individual regret bounds for the distributed online alternating direction method of multipliers. IEEE Transactions on Automatic Control, 64(4), 1746-1752.
>
> [2] Koppel, A., Jakubiec, F. Y., \& Ribeiro, A. (2015). A saddle point algorithm for networked online convex optimization. IEEE Transactions on Signal Processing, 63(19), 5149-5164.
>
> ```
> Q3: There are specific corrections of DGD that close the gap between it and approaches based on duality, which are also overlooked.
> ```
>
> Our goal is to design provably no-regret communication-efficient distributed online algorithms. We begin with the GD-based algorithm with compressors, which is fundamental. We find the key difference in algorithm design for online optimization when compressed communication is applied. After that, we also extend our GD-based algorithm (DC-DOGD) to the bandit-feedback settings and propose gradient-free online algorithms (DC-DOBD, DC-DO2BD). We are the first to give no-regret guarantees under compressed communication. We believe that this paper is an important step in this direction, and in the future we can design more distributed online algorithms with compressed communication based on more powful methods, such as EXTRA, NIDS, and so on.
>
> ```
> Q4: There are much more related works than [7]-[9] in the study of decentralized online convex optimization. Moreover, references [8] and [9] appear to be about constrained settings which are not particularly pertinent here.
> ```
> Thank you for reminding me. We will enrich related works and only keep the most relevant references in the camera-ready version.

---

> > ### Comment · Reviewer_wakL · 2022-08-07
> > **On the Author's Rebuttal**
> >
> > The reply to Question 1 is invalid. The authors state "there has been no work on provable online distributed communication-efficient ADMM or primal-dual method, to the best of our knowledge" but a quick Google search reveals:
> >
> > Zhou, S., & Li, G. Y. (2021). Communication-efficient ADMM-based federated learning. arXiv preprint arXiv:2110.15318.
> >
> > ben Issaid, Chaouki, Anis Elgabli, and Mehdi Bennis. "Local Stochastic ADMM for Communication-Efficient Distributed Learning." 2022 IEEE Wireless Communications and Networking Conference (WCNC). IEEE, 2022.
> >
> > Ma, C., Jaggi, M., Curtis, F. E., Srebro, N., & Takáč, M. (2021). An accelerated communication-efficient primal-dual optimization framework for structured machine learning. Optimization Methods and Software, 36(1), 20-44.
> >
> > Lan, Guanghui, Soomin Lee, and Yi Zhou. "Communication-efficient algorithms for decentralized and stochastic optimization." Mathematical Programming 180.1 (2020): 237-284.
> >
> > This suggests that the authors are not doing their due diligence in terms of literature review. There is also no quantitative contrast between communication efficiency in terms of some coordination scheme as compared with randomized asynchronous updates.
> >
> > The response to question 2 is sufficient, in the sense that the authors are correct that the numerical performance of duality-based approaches is not reflected in the similarity of the regret bounds between these approaches and DGD. I suspect this is because both algorithms are effectively operating with diminishing step-size. In the constant step-size regime, the contrast in terms of constraint violation would be more apparent.
> >
> >
> > The response to question 3 misunderstands the point. Linear convergence can be achieved for primal-dual/ADMM methods in the offline setting, and the way to achieve this for DGD is with a specific correct as detailed in EXTRA. The online analogue of linear convergence is logarithmic regret. Can the authors comment on when communication-efficient DGD could achieve logarithmic regret, or what fundamental limitations may preclude this? I suppose that an EXTRA-type algorithm would be required to achieve this. Note: I am referring to this paper:
> >
> > Shi, W., Ling, Q., Wu, G., & Yin, W. (2015). Extra: An exact first-order algorithm for decentralized consensus optimization. SIAM Journal on Optimization, 25(2), 944-966.
> >
> > The response to question 4 is sufficient.

---

> > > ### Author Response · Authors · 2022-08-08
> > > **On the Reviewer's reply**
> > >
> > > Thanks for your reply.
> > > ```
> > > Point 1: The authors state "there has been no work on provable online distributed communication-efficient ADMM or primal-dual method, to the best of our knowledge" but a quick Google search reveals. There is also no quantitative contrast between communication efficiency in terms of some coordination scheme as compared with randomized asynchronous updates.
> > > ```
> > > We suppose that the reviewer may have overlooked that the problem we consider is an **online** problem. The four papers that the reviewer just mentioned are **not** about **online** optimization, but offline optimization. As we stated in the rebuttal, we acknowledged that there had been either **communication-efficient** versions or **online** versions of ADMM/primal-dual methods, but there had been no work on the combination of them, that is, **online distributed communication-efficient** ADMM/primal-dual methods with **no-regret guarantees**. It is non-trivial to extend the communication-efficient offline ADMM to the **online** setting with regret analysis.
> > >
> > > In our paper, the communication efficiency is reflected by the transmitted bits reduction. For example, Fig. 2b shows that DC-DOGD with compression ratio  $\omega=0.05$ has approximately 10$\times$ reduction on transmitted bits to reach certain average regret compared with DAOL. We will consider more detailed quantitative descriptions of the communication efficiency in the future.
> > >
> > > ```
> > > Point 3: Can the authors comment on when communication-efficient DGD could achieve logarithmic regret, or what fundamental limitations may preclude this? I suppose that an EXTRA-type algorithm would be required to achieve this.
> > > ```
> > > Our communication-efficient DGD (DC-DOGD) could achieve logarithmic regret when local loss functions are strongly-convex with bounded gradients (Theorem 1), which are mild and common assumptions in online optimization. We do not need EXTRA-type algorithms to achieve this.
> > > Besides, even with the bandit information feedback, our two-bandit gradient-free algorithm (DC-D2OBD) could also achieve logarithmic regret when local loss functions are strongly-convex and $l$-Lipschitz continuous (Theorem 3). Note that in offline optimization, the linear convergence of EXTRA also needs the (restricted) strongly-convex assumption, while on the overall objective function.
> > >
> > > In addition, we wonder whether or not your concerns of Q5-Q8 have been addressed (in Part 2/2). Please let us know if you still have any unclear parts of our work.

---

### Official Review · Reviewer_d9SB · 2022-07-10

**Rating:** 5
**Confidence:** 1
**Soundness:** 4 excellent
**Presentation:** 3 good
**Contribution:** 4 excellent

**Summary:**

The paper proposes compressed distributed online convex optimization algorithms with full information feedback, one-point, and two-points bandit feedback.

**Questions:**

Unfortunately, the paper is out of my area of research and expertise.

**Limitations:**

No limitations are found.

**Strengths And Weaknesses:**

The paper proves the convergence of the proposed methods and provides computational evidence for the methods.

---

> ### Author Response · Authors · 2022-08-02
> **Response to the reviewer d9SB**
>
> Thank you for your review and positive rating. Although this paper is out of your area of research, we hope that the explanation below can help you have a better understanding of our work.
> 1. Key novelty. We are the first to give provably no-regret distributed online algorithms that work with compressors. We believe that this paper is an important step in this direction.
> 2. Possible applications. Distributed online optimization can be used in a) distributed training for large-scale machine learning tasks, such as spam filtering, dictionary learning, ad. selection, and so on; b) decentralized learning tasks, such as decentralized target tracking, formation control, and so on. Our algorithms address the communication bottleneck and can effectively reduce the total transmitted bits for distributed online training.

---

### Official Review · Reviewer_rpWg · 2022-07-11

**Rating:** 5
**Confidence:** 3
**Soundness:** 3 good
**Presentation:** 3 good
**Contribution:** 3 good

**Summary:**

The paper applies the data compression scheme to the GD-based distributed online algorithms. Three algorithms are proposed and analyzed with difference-compressed communication schemes. Explicit regret bounds are derived in closed form.

**Questions:**

1. Distributed convex optimization with compressed communication has been recently studied. What is the key difference/novelty in algorithm design for online optimization problems when compressed communication is applied?

2. Without compressed communication, distributed convex online optimization has been well studied for time-varying graphs. Is it hard to extend from a fixed graph to a time-varying graph for the current algorithms?

3. Stepsizes in the paper are designed with constant parameters given in the assumptions. It is usually not needed for standard single agent cases, nor distributed convex (online) optimization without compressed communication. Such parameter involvement requires the machines know more about the underlying problem, which is sometimes unavailable. Is this because of the purpose of analysis?

**Strengths And Weaknesses:**

The idea of the paper is simple, while the algorithm design and analysis are complicated. The paper is well written.

The paper claims that "algorithm design and theoretical principle are challenging since the compression error, projection error, and consensus error will be coupled", which is not very clear yet. For example, distributed convex optimization with compressed communication has been studied recent years; what is the challenge when moving from convex optimization to convex online optimization with compressed communication?

---

> ### Author Response · Authors · 2022-08-02
> **Response to the reviewer rpWg (Part 2/2)**
>
> ```
> Q3: The stepsizes designed in the theorems
> ```
> The stepsizes given in the theorems are designed for the purpose of analysis. The consensus stepsize $\gamma$ in Eq. (3) is designed to ensure the spectrum radius of the matrix $U(\gamma)$ (defined in Lemma 3 in Appendix) less than 1, so as to estimate the consensus error $\mathbb E_Q\\|X^t-\bar X^t\\|^2$ in $e_t$ (see Lemma 4 and Eq. (65)). The gradient descent stepsize $\eta_t$ is designed for three purposes: estimating $e_t$, minimizing the regret bounds, and simplifying the regret bounds. For example, in Theorem 1, the gradient descent stepsize for the convex case is designed as $\eta_t=\frac{D}{G\sqrt{t+c}}$ with a constant $c\geq \frac{8}{3\gamma\delta}$, where $c$ is chosen to make $e_t$ be estimated (see Lemma 4), $\frac{1}{\sqrt t}$ ensures $\mathcal O(\sqrt T)$ regret bound, and $\frac{D}{G}$ is chosen to simplify the regret bound such as (in line 584 in Appendix):
> $$\mathbb E_Q\operatorname R(j,T) \leq\frac{N\boldsymbol D^2}{2}\frac{G\sqrt{T+c}}{\boldsymbol D}+4\sqrt 3\left(\sqrt N+\frac{2\sqrt 3}{\gamma\delta}+1\right)\left(1+\frac{1}{\gamma\delta}+\frac{1}{\omega}\right)N\boldsymbol G^2\frac{2D}{\boldsymbol G}\sqrt{T+c}.$$
> The addition of $c$ is common in distributed algorithm design with compressor [14,15], and the addition of coefficient (such as $\frac{D}{G}$) is common in distributed online algorithm design [7,18].
>
> Since the stepsizes chosen in the Theorems is a sufficient but unnecessary condition for no-regret, in practice we can pick appropriate stepsizes by tuning.

---

> ### Author Response · Authors · 2022-08-02
> **Response to the reviewer rpWg (Part 1/2)**
>
> Thank you for the review. We hope that the below will help in clarifying your concerns.
> ```
> Q1: The key difference/novelty in algorithm design for online optimization problems when compressed communication is applied
> ```
> There are two key points to design provably no-regret distributed online algorithms that work with compressors.
> 1. Let the noise introduced by the compression vanish. Note that directly compressing the local state variable and spreading it will lead to the algorithm oscillation or even divergence. By allowing nodes to add replicas of neighboring states and compress the state-difference, we can address this problem. The replica $\hat x_i^{t+1}$ actually track the true state $x_i^t$. Then comes the next problem:
> 2. Bound the regret. If we first do the gradient descent, followed with the difference-compressed communication and finally the gossip consensus, as [14] and [18] did, then the regrets will be really difficult to estimate.
> Technically, in the traditional proof scheme, $\sum_{j=1}^Na_{ij}x_i^t\in\mathcal K$ since $x_i^t\in\mathcal K$ and $\sum_{j=1}^Na_{ij}=1$, while we do not have $\sum_{j=1}^Na_{ij}\hat x_i^{t+1}$ in $\mathcal K$ although $\hat x_i^{t+1}$ tracks $x_i^t$. This small difference results in big difficulty in the projection error estimation, which is essential for regret analysis. We solve this by carefully adjusting the algorithm design. We first do the difference-compressed communication (steps 2-4), and then do the gossip consensus and gradient descent simultaneously (step 5). By the projection property, we can carry out the estimation in Line 530 in Appendix, which successfully connects the projection error with the tracking error and stepsize. Besides, there are many tricks such as designing proper consensus stepsize to control the overall errors. One of the strengths of this paper is the proof novelty.
>
> ```
> Q2: Is it hard to extend from a fixed graph to a time-varying graph for the current algorithms?
> ```
> We can extend our work from a fixed graph to a time-varying graph. We give the following three cases.
>
> 1. Extend to the **random switching undirected networks**, such as the well-known Erdos-Renyi random graph model. At time $t$, an Erdos-Renyi random graph $\mathcal G_t=(\mathcal V,\mathcal E_t)$ is generated over the prescribed graph $\mathcal G$, where $\\{i,j\\}\in\mathcal E_t$ with a probability $0<p<1$ for all $\\{i,j\\}\in\mathcal E$. This setting is equivalent to performing random gossip over the graph $\mathcal G$, that is, transmiting information with the probability $p$, which satisfies Assumption 2 with $\omega=p$. Thus, our algorithm can be directly applied to the time-varying Erdos-Renyi random networks.
>
> 2. Extend to the **deterministic switching undirected networks**. Our algorithms can be applied to this setting by changing the connectivity matrix from $A$ to $A(t)$, while the difficulty lies in the regret analysis. Generally, switching networks setting assumes that the  sequence $\mathcal G_t=(\mathcal V,\mathcal E_t)$ is uniformly $B$-strongly-connected, i.e., for each $k\in \mathbb N$, the graph with edge set $\bigcup_{t=kB+1}^{(k+1)B}\mathcal E_t$ is strongly connected. The key is to estimate the average consensus error together with the noise introduced by compression over the $B$ interval, and show it declines exponentially.
>
> 3. Extend to the **time-varying directed graphs**. We can extend our DC-DOGD algorithm with the push-sum technology as follows:
> >  **Algorithm: Distributed Online Push-sum with Difference Compression (DC-DOPS)**
> > Given the time-varying connectivity matrix $A(t)$ where $a_{ij}(t)=\frac{1}{d_j^{out}(t)}$ for $j\in\mathcal N_i^{in}(t)$ and $a_{ij}(t)=0$ otherwise. Each node initializes $x_i^1=0, \hat x_i^1=0, s_i^1=0, w_i^1=1$. For $t=1$ to $T$ do in parallel for each $i\in\mathcal{V}$,
> > 1. Compress the difference vector $q_i^t=Q(x_i^{t}-\hat{x}_i^t)$ and update the replicas $\hat x_i^{t+1}=\hat x_i^t+q_i^t$
> > 2. Send $q_{i}^t$ and $w_i^t$ to all its out-neighbors, and receive $q_j^t$ and $w_j^t$ from all its in-neighbors $j\in\mathcal N_i^{in}(t)$. Update  $s_i^{t+1}=s_i^t+\sum\nolimits_{j\in\mathcal N_i^{in}(t)}a_{ij}q_j^t$ and $w_i^{t+1}=\sum\nolimits_{j\in\mathcal N_i^{in}(t)}a_{ij}w_{j}^t$
> > 3. Calculate $x_i^{t+\frac{1}{2}}=x_i^t+\gamma(s_i^{t+1}-\hat x_i^{t+1})$, $z_i^{t+1}=\frac{x_i^{t+\frac{1}{2}}}{w_i^{t+1}}$, and the local gradient $g_i^t=\nabla f_i^t(z_i^{t+1})$
> > 4. Update $x_i^{t+1} = P_{\mathcal{K}}\left(x_i^{t+\frac{1}{2}}-\eta_t g_i^t\right)$
> > Output: $\\{z_i^t\\}_{t=1}^T$
>
> To extend the regret analysis of the undirected graph setting to the directed graph setting, we need further estimate $\\|z_i^t-\bar x^t\\|$. Note that $z_i^t$ here plays the role of $x_i^t$ in the undirected graph setting. The key technique to estimate the compression error is the same. Then, along the proof scheme in our paper, the regret for DC-DOPS is likely to be estimated.

---

> ### Author Response · Authors · 2022-08-08
> **Looking forward to hearing from you**
>
> Respected reviewer rpWg,
>
> We sincerely thank you for the review and comments. We cherish this opportunity to discuss with you. Please let us know if your concerns have been addressed by our response. We would be glad to answer any further questions that you have.
>
> Many thanks for your time!
>
> Best,
>
> Authors of Paper10883

---

### Official Review · Reviewer_FBzg · 2022-07-14

**Rating:** 6
**Confidence:** 4
**Soundness:** 3 good
**Presentation:** 3 good
**Contribution:** 2 fair

**Summary:**

Paper considers distributed online convex optimization over an N-agent network. In particular, the authors consider three settings: full-feedback, where agents have access to the loss function, and (single- and two-point) bandit feedback, where agents only possess the values of the loss at points around the decision.

This work extends the DAOL method to accommodate compressed communication, and provides the first no-regret guarantees for w-contracted compressed distributed online optimization.

Gossip occurs over undirected networks with doubly-stochastic mixing. Numerical results provided for the diabetes detection dataset, using a simple online logistic-regressor with l2-regularization.

**Questions:**

* Should certainly cite [1] when discussing difference compression schemes for distributed optimization. Also note that randomized gossip is typically used to refer to the classic pair-wise averaging with random edge activation of Boyd et al. [2].

* The numerical results are interesting, but do not directly convey the advantage of decentralized learning, since fully-connected topologies still achieve the best regret with respect to the number of transmitted bits. It was shown in [3] that time-varying directed graphs can be used to improve the mixing speed of decentralized optimization algorithms over undirected graphs. Please discuss extensions to this setting, and I would be curious if push-sum based gossip (column stochastic) could be used instead of the doubly stochastic mixing to improve the numerical results in Figure 3.

[1] Karimireddy et al., Error Feedback Fixes SignSGD and other Gradient Compression Schemes, ICML 2019.
[2] Boyd et al., Randomized gossip algorithms, IEEE transactions on information theory 2006.
[3] Assran et al., Stochastic Gradient Push for Distributed Deep Learning, ICML 2019.

**Limitations:**

Limitations are addressed in the conclusion (namely the dependence on the compression ratio in the regret bounds).

**Strengths And Weaknesses:**

Strengths:
Originality & Significance: while the proposed method is highly incremental, the analysis in this work is sufficiently original, and to the best of my knowledge, provides the first no-regret guarantees for w-contracted compressed distributed online optimizers.

Clarity & Quality: the work is clear, well written, and technically sound. Additionally, numerical results support theoretical intuition (i.e., that full-feedback and double-point bandit feedback have similar regrets, which are significantly better than single-point banding feedback). Additionally, advantage of proposed method compared to exact communication DAOL is well-established, improved convergence with respect to transmitted bits, at the cost of slightly worse time-horizon convergence.

Weaknesses:
Incremental algorithm: Method is an extension of DAOL to allow for compressed communication using standard compression schemes. Moreover, the method for dealing with compression is directly borrowed from the error-feedback framework.

Assumptions: For the full-feedback setting: requires the bounded gradient assumption in the convex setting (however this can be eliminated it for strongly-convex objectives).

Unfortunately, the experiments do not directly demonstrate the advantage of sparse communication topologies on the convergence speed, as the fully-connected network in Figure 3 still exhibits the best regret with respect to the transmitted bits.

---

> ### Author Response · Authors · 2022-08-02
> **Response to the reviewer FBzg (Part 2/2)**
>
> ```
> Q4: Randomized gossip is typically used to refer to the classic pair-wise averaging with random edge activation of Boyd et al. [2]
> ```
> Thank you very much for pointing out this. To distinguish from the proper noun 'randomized gossip', we will use the term 'random gossip' in our revision to refer to the weighted averaging of neighbors with random edge activation, which is the strategy that we and [14] consider.
> ```
> Q5: Please discuss extensions to the time-varying directed graphs. If push-sum based gossip (column stochastic) could be used instead of the doubly stochastic mixing to improve the numerical results in Figure 3.
> ```
> For the time-varying directed graphs setting, we can extend our DC-DOGD algorithm with the push-sum technology as follows:
> >  **Algorithm: Distributed Online Push-sum with Difference Compression (DC-DOPS)**
> > Given the time-varying connectivity matrix $A(t)$ where $a_{ij}(t)=\frac{1}{d_j^{out}(t)}$ for $j\in\mathcal N_i^{in}(t)$ and $a_{ij}(t)=0$ otherwise. Each node initializes $x_i^1=0, \hat x_i^1=0, s_i^1=0, w_i^1=1$. For $t=1$ to $T$ do in parallel for each $i\in\mathcal{V}$,
> > 1. Compress the difference vector $q_i^t=Q(x_i^{t}-\hat{x}_i^t)$ and update the replicas $\hat x_i^{t+1}=\hat x_i^t+q_i^t$
> > 2. Send $q_{i}^t$ and $w_i^t$ to all its out-neighbors, and receive $q_j^t$ and $w_j^t$ from all its in-neighbors $j\in\mathcal N_i^{in}(t)$. Update  $s_i^{t+1}=s_i^t+\sum\nolimits_{j\in\mathcal N_i^{in}(t)}a_{ij}q_j^t$ and $w_i^{t+1}=\sum\nolimits_{j\in\mathcal N_i^{in}(t)}a_{ij}w_{j}^t$
> > 3. Calculate $x_i^{t+\frac{1}{2}}=x_i^t+\gamma(s_i^{t+1}-\hat x_i^{t+1})$, $z_i^{t+1}=\frac{x_i^{t+\frac{1}{2}}}{w_i^{t+1}}$, and the local gradient $g_i^t=\nabla f_i^t(z_i^{t+1})$
> > 4. Update $x_i^{t+1} = P_{\mathcal{K}}\left(x_i^{t+\frac{1}{2}}-\eta_t g_i^t\right)$
> > Output: $\\{z_i^t\\}_{t=1}^T$
>
> We compare the performance of DC-DOPS on the *one-peer-per-node directed time-varying graph* with DC-DOGD on the *full-connected undirected time-invariant graph*. We plot the time-averaged maximum regret versus the iterations, the running time, and the total transmitted bits, respectively, as shown in (https://github.com/happy-math/CC-DOCO/blob/main/figures/Directed_vs_Undirected.jpg). The result shows that the full-connected graph achieves fewer iterations, while the time-varying directed graph improves the online learning speed and needs fewer transmitted bits.
>
> To extend the regret analysis of the undirected graph setting to the time-varying directed graph setting, we need further estimate 1) $\\|z_i^t-\bar x ^t\\|$, where $z_i^t$ plays the role of $x_i^t$ in the undirected graph setting; 2) the average consensus error together with the noise introduced by compression over a certain time interval, which is expected to decline exponentially. The key technique to estimate the compression error is the same. Then, along the proof scheme in our paper, the regret for DC-DOPS is likely to be estimated.

---

> ### Author Response · Authors · 2022-08-02
> **Response to the reviewer FBzg (Part 1/2)**
>
> Thank you for your review and thoughtful comments. Below, we address your concerns and questions individually.
> ```
> Q1: The method for dealing with compression is directly borrowed from the error-feedback framework. Should certainly cite Karimireddy et al. [1] when discussing difference compression schemes for distributed optimization.
> ```
> We use the difference compression framework to deal with the compression in distributed online optimization. The difference compression framework (DC) is different from the error-feedback framework (EF) in what to be compressed, which results in different application fields. DC compresses the difference between the current variable and the replica variable, which is widely used in distributed optimization, while EF compresses the sum of the gradient and the residual error, which is widely used in federated learning. The insight is that successful designs have to compress something that goes to zero, otherwise the noise introduced by the compression will not vanish, which leads to the algorithm oscillation or even divergence. We give the following detailed explanation.
>
> Suppose that node A is going to compress a time-varying information $A_t$ and send it to node B. DC lets node B keep a replica of A as $\hat A_t$. A send the difference-compressed information $q_t=Q(A_{t+1}-\hat A_t)$ to B, and B updates the replica $\hat A_{t+1}=\hat A_t+Q(A_{t+1}-\hat A_t)$. Thus, $\hat A_{t+1}$ tracks $A_{t+1}$. EF lets node A keep a residual error $e_t$. A sends the error-feedback information $q_t=Q(A_t+e_t)$ to B. The rest is added to $e_t$ to be transmitted later, that is, A updates the residual error $e_{t+1}=A_t+e_t-q_t$. Eventually, all the information is transmitted, albeit with a delay. To use DC, $A_{t+1}-\hat A_t$ should go to zero; to use EF, $A_t+e_t$ should go to zero.
>
> In distributed optimization (cf. [13-16]), there is no central node, and each node collects the state information of its neighbors. Here $A_t$ is to be the state variable $x_i^t$, whose limit is nonzero in general. When all nodes are reaching a consensus optimal state, the updates of local states are small, and the differences between the replicas and the true states are also small. In other words, $A_{t+1}-\hat A_t\to0$, as $t\to\infty$. Then the compression errors are expected to vanish. Thus, DC is suitable for distributed optimization. In federated learning (cf. [32-34] in our paper and [1] as you have mentioned), there is a central node to collect gradient information from other nodes to update the global state variable $x^t$. Here $A_t$ is to be the gradient $\nabla f_i^t(x^t)$, which is expect to go to zero as $x^t$ reaches the optimal point. Besides, the rests will be sent eventually, that is, the residual error $e_t\to0$, as $t\to\infty$. Thus, EF is suitable for federated learning.
> ```
> Q2: The bounded gradient assumption
> ```
> To make it clear, we recall the assumptions in Section 2 (Full Information Feedback setting) as follows (Assumption 4 is split into 4.1 and 4.2 two parts):
>
> - Assumption 3: $\mathcal K$ is bounded.
>
> - Assumption 4.1: $f_i^t$ is convex and differentiable.
>
> - Assumption 4.2: $\nabla f_i^t$ is bouned.
>
> - Assumption 5: $f_i^t$ is strongly-convex.
>
> In both convex and strongly-convex settings, we need gradients to be bounded, which is common in online optimization [18]. In the convex setting, actually the bounded gradient assumption 4.2 can be guaranteed by assumptions 3 and 4.1. In the strongly-convex setting, to emphasize that assumption 3 is not needed here, we require assumption 4.2. In other words, if we keep assumption 3 holding in all settings, then the bounded gradient assumption 4.2 can be eliminated.
> ```
> Q3: The experiments do not directly demonstrate the advantage of sparse communication topologies on the convergence speed, as the fully-connected network in Figure 3 still exhibits the best regret with respect to the transmitted bits.
> ```
> The total number of transmitted bits is formed by the transmitted bits in each iteration and the total iterations. Sparser communication topologies with fewer edges need fewer bits to transmit information in each iteration, while more iterations are needed for decision consensus. Thus, there will be a tradeoff. In our experiments, the performances of the ring graph, the random generated graph $\mathcal G(N,2N)$, and the fully-connected graph are closs w.r.t. the transmitted bits, while $\mathcal G(N,2N)$ performs slightly better than the fully-connected graph.
>
> The network topology is involved with the parameters $\delta$ and $\beta$, which influence the choice of the consensus stepsize $\gamma$ as well as the algorithm performance. In the future, we can deeply investigate the best choice of the network topology in the sense of minimizing the total transmitted bits. Thanks for your advice.

---

> ### Author Response · Authors · 2022-08-08
> **Looking forward to hearing from you**
>
> Respected reviewer FBzg,
>
> Thank you for the precious review time. We cherish this opportunity to discuss with you. We are wondering whether your concerns have been well addressed. If you have any additional questions, it would be great to let us know. We are glad to answer them.
>
> Many thanks for your time!
>
> Best,
>
> Authors of Paper10883

---

### Meta-Review · Area_Chair_P8Jt · 2022-08-24

**Recommendation:** Accept
**Confidence:** Less certain

**Metareview:**

This paper studies a decentralized online learning algorithm that uses compressed communication and achieves the same regret (order-wise) as an analogous method which would not use any compressed communication. Several reviewers noted that the idea and formulation are straightforward, but the analysis is technically involved. It is this analysis that constitutes the main contribution of this work. After discussion with the reviewers, I am of the opinion that the contributions of this paper are sufficiently relevant and the work contains sufficient technical novelty to justify acceptance to NeurIPS.

The responses to reviewers helped to clarify several points of concern in the initial reviews. Please make sure to revise your paper for the camera ready deadline taking these suggestions into account. In particular,
* Several reviewers noted relevant related work that ought to be cited
* Clarify the differences with the error-feedback framework
* Limitations and future work regarding how sparsity of the topology impacts overall communication overhead
* Discuss potential extensions to time-varying graphs

**Award:**

No

---

### Decision · Program_Chairs · 2022-09-14

Accept